# Artificial Intelligence Predictive Models of Response to Cytotoxic Chemotherapy Alone or Combined to Targeted Therapy for Metastatic Colorectal Cancer Patients: A Systematic Review and Meta-Analysis

**DOI:** 10.3390/cancers14164012

**Published:** 2022-08-19

**Authors:** Valentina Russo, Eleonora Lallo, Armelle Munnia, Miriana Spedicato, Luca Messerini, Romina D’Aurizio, Elia Giuseppe Ceroni, Giulia Brunelli, Antonio Galvano, Antonio Russo, Ida Landini, Stefania Nobili, Marcello Ceppi, Marco Bruzzone, Fabio Cianchi, Fabio Staderini, Mario Roselli, Silvia Riondino, Patrizia Ferroni, Fiorella Guadagni, Enrico Mini, Marco Peluso

**Affiliations:** 1Research and Development Branch, Regional Cancer Prevention Laboratory, ISPRO-Study, Prevention and Oncology Network Institute, 50139 Florence, Italy; 2Department of Experimental and Clinical Medicine, University of Florence, 50134 Florence, Italy; 3Institute of Informatics and Telematics, National Research Council, 56124 Pisa, Italy; 4Department of Surgical, Oncological and Oral Sciences, University of Palermo, 90127 Palermo, Italy; 5Department of Health Sciences, University of Florence, 50139 Florence, Italy; 6Department of Neurosciences, Imaging and Clinical Sciences, “G. D’Annunzio” Chieti-Pescara, 66100 Chieti, Italy; 7Clinical Epidemiology Unit, IRCCS-Ospedale Policlinico San Martino, 16131 Genova, Italy; 8Medical Oncology Unit, Department of Systems Medicine, Tor Vergata University, 00133 Rome, Italy; 9BioBIM (InterInstitutional Multidisciplinary Biobank), IRCCS San Raffaele Roma, 00166 Rome, Italy; 10Department of Human Sciences & Quality of Life Promotion, San Raffaele Roma Open University, 00166 Rome, Italy

**Keywords:** artificial intelligence, chemotherapy, targeted therapy, responders, radiomics, biomarkers, algorithm, colorectal cancer metastasis

## Abstract

**Simple Summary:**

Metastatic colorectal cancer (mCRC) has high incidence and mortality. Nevertheless, innovative biomarkers have been developed for predicting the response to therapy. We have examined the ability of learning methods to build prognostic and predictive models to predict response to chemotherapy, alone or combined with targeted therapy in mCRC patients, by targeting specific narrative publications. After a literature search, 26 original articles met inclusion and exclusion criteria and were included in the study. We showed that all investigations conducted in this field provided generally promising results in predicting the response to therapy or toxic side-effects, using a meta-analytic approach. We found that radiomics and molecular biomarker signatures were able to discriminate response vs. non-response by correctly identifying up to 99% of mCRC patients who were responders and up to 100% of patients who were non-responders. Our study supports the use of computer science for developing personalized treatment decision processes for mCRC patients.

**Abstract:**

Tailored treatments for metastatic colorectal cancer (mCRC) have not yet completely evolved due to the variety in response to drugs. Therefore, artificial intelligence has been recently used to develop prognostic and predictive models of treatment response (either activity/efficacy or toxicity) to aid in clinical decision making. In this systematic review, we have examined the ability of learning methods to predict response to chemotherapy alone or combined with targeted therapy in mCRC patients by targeting specific narrative publications in Medline up to April 2022 to identify appropriate original scientific articles. After the literature search, 26 original articles met inclusion and exclusion criteria and were included in the study. Our results show that all investigations conducted on this field have provided generally promising results in predicting the response to therapy or toxic side-effects. By a meta-analytic approach we found that the overall weighted means of the area under the receiver operating characteristic (ROC) curve (AUC) were 0.90, 95% C.I. 0.80–0.95 and 0.83, 95% C.I. 0.74–0.89 in training and validation sets, respectively, indicating a good classification performance in discriminating response vs. non-response. The calculation of overall HR indicates that learning models have strong ability to predict improved survival. Lastly, the delta-radiomics and the 74 gene signatures were able to discriminate response vs. non-response by correctly identifying up to 99% of mCRC patients who were responders and up to 100% of patients who were non-responders. Specifically, when we evaluated the predictive models with tests reaching 80% sensitivity (SE) and 90% specificity (SP), the delta radiomics showed an SE of 99% and an SP of 94% in the training set and an SE of 85% and SP of 92 in the test set, whereas for the 74 gene signatures the SE was 97.6% and the SP 100% in the training set.

## 1. Introduction

Colorectal cancer (CRC) is one of the most common cancers and a main leading cause of cancer death in the world [1,2]. The most recent data from GLOBOCAN recorded more than 1.88 million cases and 916,000 deaths per year [3]. Despite the effort of screening programs for early detection of CRC, about 20% of CRC patients present with metastatic disease at diagnosis (synchronous metastases), with low curative surgical control and high rate of tumor-related deaths [4]. In addition, about 25% of patients who present with localized disease will develop metastases at a later time (metachronous metastases) even after radical loco-regional treatment (surgery, radiotherapy) of the primary tumor and adjuvant chemotherapy [4]. When localized, the vast majority of CRC patients are curable by surgery (stage I and low-risk stage II), but the prognosis for metastatic CRC (mCRC) patients remains poor. Indeed, while the the 5-year overall survival (OS) rate for localized CRC is approximately 90%, this estimate drastically decreases for patients with metastatic CRC (mCRC) to a mere 15.5% (https://seer.cancer.gov/statfacts/html/colorect.html (accessed on 30 April 2022)). Treatment strategies are commonly based on clinical staging, with surgery, neoadjuvant (either with chemotherapy or radiation therapy), and postoperative adjuvant chemotherapy for high-risk locally advanced stages (high-risk stage II and stage III) and with systemic anticancer drugs as primary treatment for metastatic disease.

A combination of cytotoxic agents including fluoropyrimidine, oxaliplatin and/or irinotecan, with a targeted agent, either an anti-epidermal growth factor receptor (EGFR) or an anti-vascular endothelial growth factor (VEGF) monoclonal antibody (MoAb), is nowadays considered the primary standard treatment for mCRC patients to achieve improved disease control and prolong survival [5,6]. Results from the first phase III clinical trial evaluating the addition of the anti-VEGF MoAb bevacizumab to 5-fluorouracil and irinotecan represented the beginning of a new era for mCRC treatment by showing that such combined treatment could result into a significant improvement in tumor objective response (OR), progression-free survival (PFS), and OS [7]. While the median OS for mCRC patients following single agent 5-fluorouracil or biochemical modulation of 5-fluorouracil by folinic acid ranged from 10 to 12 months in the 1980s, it has now been significantly increased, to approximately 26–33 months, by the introduction of targeted therapy [8]. Despite these treatment advances, therapeutic outcomes are still unsatisfactory and variable among patients receiving the same drugs [9], due to the occurrence of tumor drug resistance either at the time of initial treatment (intrinsic resistance) or after the first-line treatment with cytotoxic and targeted agents (acquired resistance) in nearly all patients [10]. Moreover, a large part of mCRC patients can experience several adverse effects, as anemia, neutropenia, nausea, vomiting, diarrhea and cardiotoxicity [11] that might cause treatment delay or even discontinuation.

Evolution of genetically and/or epigenetically diverse tumor-cell populations during tumor growth and progression is considered the main factor causing tumor heterogeneity, displaying inherent functional variability in tumor propagation potential and tolerance to pharmacological treatment with either cytotoxic or targeted cancer therapeutics [5]. Personalized treatments for mCRC disease are today limited to a small number of drugs towards molecular targets such as anti-VEGF, anti-EGFR for *RAS* wild-type tumors, encorafenib for *BRAF* (*V600E*) mutated tumors, programmed death-ligand 1/programmed cell death-1(PDL-1/PD-1) inhibitors for mismatch repair deficiency (dMMR)/microsatellite instability-high (MSI-H) tumors, and possibly *KRAS* tyrosine kinase inhibitors for G12C *KRAS* mutated tumors [12]. Recent additions to the mCRC armamentarium are the inhibitors of HER2 (rarely overexpressed in CRC, with a higher prevalence in *RAS*/*BRAF*–wild type tumors) and *NTRK* gene fusions (limited to cancers that are wild-type for *KRAS*, *NRAS*, and *BRAF*) [13,14]. As above reported, although VEGF and its receptors (VEGFR) inhibitors have a therapeutic role in the treatment of mCRC [6] resulting in angiogenic blockade, their use still lacks stratification based on predictive biomarkers of drug response of clinical relevance, especially in terms of resistance [15]. The use of intensive chemotherapy combination regimens requires also a fine balance between clinical outcome and treatment related toxicity. Therefore, there is an unmet need for early identification of mCRC cases who will be responsive to specific regimens and clinical indicators of toxicity burden. Indeed, at the present, no imaging criteria or molecular biomarkers are available in clinical practice today for early identification of treatment response before the start of the therapy and even the radiologists cannot have sufficient imaging information on the baseline examination to predict which tumor lesions will respond to the treatments.

Artificial intelligence (AI), a field of computer science that mimics human intelligent behavior and performs tasks that commonly require human intervention, has gained interest in the field of cancer research for its problem solving, decision making and pattern recognition abilities [16]. Machine learning (ML) and deep learning (DL), two subsets of AI, have emerged as important tools that could effectively improve the field of molecular cancer characterization, from epidemiology [17], to diagnosis, prognosis and patient classification [18], with higher performance than traditional approaches [19,20,21,22,23,24]. Accumulating evidence suggests that radiomics can be used to study tumor heterogeneity and to predict therapy in CRC [25]. Biological insight on data-driven radiomics features has been recently provided by different biological metrics such as gene expression, immunohistochemistry staining intensity and microscopic histologic textures [26]. In particular, radiomics in combination with molecular biomarkers has even been shown to be capable of predicting genetic mutations including both *KRAS* mutations and *KRAS*/*BRAF* status in CRC patients [25]. Nevertheless, there are only scattered reports on applying AI techniques to predict treatment response in oncology, especially for metastatic colorectal disease; therefore, a systematic review and meta-analysis was designed to assess the effectiveness of learning algorithms to predict the response to chemotherapy alone or combined to targeted therapy, as well as the individual toxicity of standard regimens in this patient setting. To the best of our knowledge, this represents the largest meta-analysis of predictive classifiers for mCRC so far, aimed at validating or refuting previous findings, with the final goal of evaluating whether learning models could be used in clinical trials to assess prognosis or toxicity in standard of care settings and as predictors of therapy response.

### 1.1. Machine and Deep Learning Techniques

In recent years, the application of computer aided intelligence has expanded to healthcare for finding novel clinical solutions drawing on the subfields of the traditional ML and the cutting-edge DL architectures and seeking improved performance in respect to traditional statistical methods [18]. They are both capable of learning from data and identifying the distribution and patterns of key features, but have different learning structures, with DL mimicking the property of the human brain for data processing using multi-layered structured models [18]. ML has emerged as a powerful computational approach that uses algorithms to iteratively and automatically learn from data to perform a task without being specifically programmed [27]. Main ML analytical tasks consist in classification, pattern recognition, clustering, key feature identification and construction of predictive models using three different approaches: (i) supervised, (ii) unsupervised, and (iii) reinforcement learning [16]. In supervised learning, labeled data are used to train algorithms to classify data or predict outcomes; while in unsupervised learning, unlabeled or uncategorized data are used in learning processes and relationships are discovered on the basis of hidden similarities; lastly, reinforcement learning is an environment-driven approach, i.e., it enables software agents to automatically receive rewards or penalties to improve their performance [27]. Supervised algorithms can be grouped in classification and regression applications, where the former refers to the prediction of categorical outcomes, and the latter to the prediction of continuous outcomes. Common supervised ML models are Support Vector Machines (SVM), Decision Trees (DT), and Artificial Neural Networks (ANN). SVM is one of the most used algorithms which map the input variables into a feature space of higher dimensionality using the kernel trick [28] and identify the best decision surface, the hyperplane, which separates the dataset into two different classes. The DTs are tree-based classification models where the dataset is split at each node of the tree using input feature thresholds and leaves represent the final classification decision. A DT performs features selection at each node using metrics as information gain to select the most informative features during its training. DTs can be used for solving regression and classification problems also by learning simple decision rules inferred from prior data in the training process [18]. ANNs are biologically inspired computational networks based on studies of the brain and nervous system, that can compute values from inputs and examine interactions between groups of representative features to predict outcomes [29]. Unsupervised algorithms analyze unlabeled datasets for clustering, density estimation, feature learning, dimensionality reduction, finding association rules and anomaly detection. Clustering is a common task in biomedical research employing molecular profiling to find similarities among samples and allow stratification of patients into clinically-meaningful subgroups. Among the available techniques, the most frequently used are hierarchical and k-means clustering. Besides clustering, Principal Component Analysis (PCA), factor analysis, or more complex techniques such as matrix factorization are frequently used techniques to address dimensionality reduction especially for datasets with large number of variables and relative small numbers of samples [30]. Semi-supervised learning methods also exist which exploit labeled and unlabeled data to train a model [27]. On the other hand, DL is a subfield of AI technique that uses multi-layered neural network algorithms similar to the neurological architecture to infer predictions [31]. DL is essentially a computational architecture composed by different processing layers, namely an input, (multiple) hidden, and an output layer [27]. In respect of ML, the neural network architecture is used for deep processing, automated feature extraction and pattern recognition along complex data, as large-scale image classification. In DL architecture, the representation of output data produced by each layer is used as input to the subsequent layer and so forth, retaining the data of interest and discarding other information. By this approach, DL can identify complex combinations of original features within combinations of deeper layers and extrapolate new features that will increase the predictive value in respect to original ones [31]. Common DL methods are the Recurrent Neural Networks (RNN) and the Convolutional Neural Networks (CNN). RNNs are designed for making predictive models based on temporal sequences using current and past features to automatically construct algorithms. CNN is a type of network that is specified for image analysis and computer vision. CNN is able to perform specific tasks such as image classification and automated features extractions [32]. In ML/DP, the quality of the datasets is fundamental and a preprocessing strategy, encompassing feature standardization and extraction is a necessary step before applying any learning algorithm.

### 1.2. Standard Chemotherapy, Targeted Therapy and Mutational Profiles

Since the Sixties, 5-fluorouracil represents the basis of mCRC treatment. In the last 25 years, capecitabine, a pro-drug of 5-fluorouracil, oxaliplatin, a platinum compound, and irinotecan, a topoisomerase I inhibitor, have also been introduced in the clinical practice leading to improved patient outcomes [33,34]. Currently, the main doublet chemotherapies used in first-line treatment are FOLFIRI (5-fluorouracil/folinic acid plus irinotecan) and FOLFOX (5-fluorouracil/folinic acid plus oxaliplatin) regimens, while CapeOX (capecitabine plus oxaliplatin) and XELIRI (capecitabine plus irinotecan) represent less commonly used chemotherapy regimens. A triple chemotherapy, the FOLFOXIRI or FOLFIRINOX (5-fluorouracil/folinic acid plus oxaliplatin and irinotecan) regimen, is also a first-line treatment option in selected cases [5,35]. The availability of anti-EGFR MoAbs, such as cetuximab or panitumumab, and of anti-VEGF MoAbs, such as bevacizumab, from the early years after 2000 have further increased activity and efficacy of first-line drug treatment in mCRC when combined with the above chemotherapy regimens [5]. These biological therapeutics can act directly on tumor cells through different mechanisms including targeting cell surface receptors for specific relevant growth factors (e.g., EGF) and consequently by blocking oncogenic signaling leading to cell proliferation arrest, inhibition of cell differentiation and cell migration, interference with cell cycle progression and enhancement of apoptotic death. They can also act indirectly on tumor cells through inhibition of blood vessels in the tumor microenvironment by targeting specific endothelial cell membrane receptors or their growth factors (e.g., VEGF) [5]. More recently, immune checkpoint inhibitors (ICIs), working by blocking immunoinhibitory signals and enabling patients to induce an anti-tumor action, have also proven active in the treatment of mCRC [36]. ICIs targeting PD-1 are in fact highly effective in a selected mCRC patient population whose tumors display dMMR or MSI-H [37]. 

In the case of anti-EGFR MoAbs, the study of mutational profiling in *KRAS* and *NRAS* sequences is highly relevant for the choice of clinical treatments [38]. Patients with colorectal metastasis bearing mutations in *KRAS* or *NRAS* are considered non-responders or poor responders to anti-EGFR therapy, whereas the presence of *BRAF* V600E mutation identifies a very poor-prognosis subgroup of patients [39], who also display resistance to anti-EGFR MoAbs [40,41]. In *BRAF* mutated tumors the use of a potent *BRAF* tyrosine kinase inhibitor, such as encorafenib, can overcome resistance to anti-EGFR MoAbs [5]. In RAS wild-type tumors, cetuximab and panitumumab in combination with chemotherapy can instead significantly improve the median OS of mCRC patients [6]. Difference in response to chemotherapy can exist among patients carrying the same mutation profiles, since patients with *KRAS* or *NRAS* wild-type tumors can be partially not responsive to anti-EGFR agents [42]. Besides the above mentioned V600E *BRAF* mutations, also *PIK3CA* mutations, PTEN loss, HER2 overexpression/amplification or MET amplification have been linked to resistance to anti-EGFR targeted therapies [43]. Interestingly, excision repair cross complementation group 1 (ERCC1) expression is inhibited by *KRAS* mutation, making tumors more sensitive to cytotoxic treatments [44]. This has led to hypotheses that several other genetic and epigenetic mechanisms can be involved in primary or secondary drug resistance and tumor progression [45].

As above reported, dMMR/MSI-H are predictive biomarkers of tumor response to ICIs pembrolizumab and nivolumab MoAbs against PD-1 [2,6]. Indeed, PD-1 blockade with pembrolizumab or nivolumab in this setting of mCRC patients has provided an efficacious treatment option for chemotherapy resistant tumors [46,47,48,49]. In this regard, later phase III clinical studies of pembrolizumab versus standard chemotherapy in patients with previously untreated MSI-H or dMMR mCRC have shown that pembrolizumab significantly improves PFS [50] and OS [51] as first-line therapy as well as health-related quality of life [37,52]. These data have supported FDA and EMA approvals of pembrolizumab as a new standard of care in this mCRC subpopulation. Indeed, a treatment based on “double immunity” with nivolumab in combination with ipilimumab has also been evaluated in a non-randomized phase II trial in first-line setting with an OR rate of 69%, and a median PFS and OS not yet reached at a 24-month follow-up [53]. The v1.2021 NCCN guidelines now recommend nivolumab ± ipilimumab or pembrolizumab as a first-line treatment for patients with unresectable advanced or mCRC with MSI-H/dMMR status (NCCN Clinical Practice Guideline in Oncology. Version 1.2021. Available at: Rectal Cancer.NCCN.org (accessed on 30 April 2022)). ICIs have not yet shown definitive evidence of clinical efficacy in patients with proficient MMR or MS-stable mCRC. However, a recent study has shown that nivolumab added to encorafenib and cetuximab in V600E *BRAF* mutated, microsatellite-stable mCRC induces higher response rates [54] when compared with encorafenib and cetuximab alone from the previously reported study [55]. Moreover, retrospective studies have shown that patients with distal primary tumors have improved outcomes as compared to those with proximal tumors [56]. Proximal tumors are more aggressive and differ from distal tumors in showing diverse molecular and pathological characters [57]. Tumor location can be also associated with the response to targeted therapy. mCRC patients with distal tumors respond better to anti-EGFR agents than those with proximal tumors [39], possibly due to the different embryologic origin of distal tumors, whether embryonic hindgut or embryonic midgut. Interestingly, differences in embryologic origin are associated to different phenotypes, with proximal tumors carrying *RAS* and *BRAF* mutations, whereas distal tumors carry *KRAS*, *APC*, *PIK3CA*, and *TP53* mutations [58]. Similarly, therapy responses can be different among these two kinds of colorectal tumors: cancer patients with left-sided tumors may be responsive to anti-EGFR targeted therapy, while those with right-sided tumors do not have the same benefit from conventional treatments [59].

In addition to somatic mutations, gene expression may contribute to tumor drug response. Through the years, different approaches have been used to identify genes whose expression levels could discriminate between responders and non-responders to pharmacological treatment in CRC. The “candidate gene” approach, widely used in the past years, has led to the identification of a number of potential biomarkers of drug response [60,61,62,63,64,65]. More recently, genes potentially predictive of drug response in CRC have also been identified and validated starting from an RNA seq transcriptomic approach [66,67]. In addition, by using unsupervised clustering methods, CRC transcriptomic data provided new molecular classifications, including the “consensus molecular subtypes” (CMS) [68] and the “cancer cell intrinsic transcriptional traits” (CRIS) [69]. Both of them, although not currently recommended in clinical practice, are good predictors of prognosis whereas their ability to predict drug response has not been proven.

## 2. Methods

This review and meta-analysis was conducted according to the Preferred Reporting Items for Reviews and Meta-Analyses (PRISMA) guidelines [70] and not registered in PROSPERO.

### 2.1. Search Strategy and Data Extraction

A systematic literature search was performed using National Library of Medicine medical literature database (PubMed, https://pubmed.ncbi.nlm.nih.gov/ (accessed on 30 April 2022)) from January 2005 up April 2022, to identify studies performed with AI techniques for the development and validation of accurate prognostic tools to be employed for predicting treatment response and/or individual toxicity in mCRC patients. This objective was obtained by targeting specific narrative publications in medical literature and identifying appropriated original scientific articles. The publications were searched with main domains connected by the Boolean operators “AND” and “OR”: “*artificial intelligence*”, “*machine learning*”, “*deep learning*”, “*neural networks*”, “*predictor*”, “*classifier*”, “*signature*” OR “*radiomics*” AND “*metastatic colorectal cancer*” OR “*colorectal cancer*” AND “*chemotherapy*”, “*targeted therapy*”, “*cytotoxic regimens*”, “*immunotherapy*”, “*response to therapy*” OR “*responders*”. To supplement PubMed search, additional articles were collected by checking reference lists of identified studies, reviews and systematic reviews. Two investigators (EL and VR) have independently screened the PUBMED publications against the criteria of inclusion and exclusion. In case of disagreement, the opinion of a third investigator (MP) was requested.

When available, the following variables were extracted for each eligible study: first line and previous chemotherapy and/or targeted therapy, AI networks (ML or DL), total sample size, training and validation cohort sample sizes, input features used to feed algorithms, signature used for learning models (radiomics or molecular biomarkers), learning model performances at training and/or at validation sets. We distinguished performance metrics based on the task, e.g., classification or regression/survival analysis, for each study. A detailed description of each metric is reported in Table 1.

### 2.2. Study Selection

Studies had to meet the following prespecified inclusion criteria: (1) patients with mCRC; (2) use of AI algorithms to assess response to therapy or side effects; and (3) published in English language. Exclusion criteria were: (i) case reports, review articles, editorials, letters, comments, and conference abstracts; (ii) studies lacking therapy response analysis; and (iii) studies lacking learning model performance evaluation.

### 2.3. Objectives

The primary objective of the study was to evaluate the effectiveness of learning algorithms in the prediction of the response to chemotherapy alone or combined to targeted therapy and/or side effects for mCRC patients. The secondary objective was to assess the efficacy of the aforementioned algorithms in discriminating responder mCRC patients/treated metastatic lesions vs. non-responders mCRC patients/untreated metastatic lesions.

### 2.4. Publication Quality

The study of publication quality was assessed by two investigators (MP and EL) using study quality items that were criteria of quality publication in the Luo scale [71], in keeping with a recent systematic review on ML studies [72].

### 2.5. Rational Framework

A rational framework for interpreting model performances was developed using the area under the receiver operating characteristic (ROC) curve (AUC), and the sensitivity (SE) and the specificity (SP) estimates reported in the AI models. AUC values were interpreted as follows: 0.6–0.7 (worthless), 0.7–0.8 (poor), 0.8–0.9 (good), and >0.9 (excellent) [73]. In addition, the SE was referred to the ability of a test to correctly identify patients who are responders, whereas the SP to the ability to correctly classify those who are non-responders [74,75].

### 2.6. Statistical Analysis

A meta-analysis was carried out including only the studies reporting the AUC or HR estimates with 95% C.I. in training and/or validation sets. The *random effects model* [76,77,78] was used to calculate global response to therapy, reported as overall AUC calculated from the receiver operating characteristic (ROC) analyses or overall HR with 95% CIs from proportional hazards models (Cox regression). The pooled HR was considered statistically significant if the 95% C.I. did not include 1.0 while the confidence intervals for global AUC not including 0.50 suggest that the test is able to discriminate between the comparison groups. To stabilize the within-study variance, the logit of the AUC and the natural logarithm of HR were used for the statistical analysis. To assess whether the pooled performances were stable or significantly dependent on one or more studies, sensitivity analyses were conducted by interactively recalculating the overall estimates after exclusion of one study at a time. Egger’s test was also applied to assess the occurrence of publication bias. All statistical analyses were conducted using Stata Software version 14.2 (StataCorp LLC, College Station, TX, USA).

## 3. Results

A total of 1947 records were identified from databases using the predefined search criteria (Figure 1). After a review of titles and abstracts, duplicates and papers not related to prediction models were rejected. 221 remaining manuscripts were analyzed in-depth to evaluate their concordance with inclusion and exclusion criteria. Other articles were collected by checking reference lists of identified studies, reviews and systematic reviews. The final database consisted of 26 studies with 50,257 patients eligible for inclusion in the database [74,75,79,80,81,82,83,84,85,86,87,88,89,90,91,92,93,94,95,96,97,98,99,100,101,102]. Table 2 and Table 3 show the 26 original AI studies that were included in the systematic review. There were 13 studies with 4193 patients, 5 of radiomics and 8 of molecular biomarkers learning models, that evaluated the response to chemotherapy alone, and 2 other studies with 36,050 patients that assessed the individual toxicity of standard regimens.

Likewise, there were 14 studies, with 3 investigations in common with Table 3, with 10,014 patients, 7 of radiomics and 7 of molecular biomarkers based classifiers that analyzed the response to targeted therapy. The AI studies were listed and divided in predictive learning models of response or side effects to chemotherapy alone and predictive learning models of response to targeted therapy. Investigations collected under each category were stratified in imaging, molecular and clinical biomarkers based models, summarized in detail and discussed. Table 2 and Table 3 report the variables extracted for each eligible study.

### 3.1. Outcomes and Performance Estimates of Predictive Models

Different clinical endpoints, including the median values of the OS, the PFS, the TTNT or the Response Evaluation Criteria in Solid Tumors (RECIST) criteria, were used to analyze the response to therapy in periods of time consistent between different studies, commonly after 6 or more months from the last cycle of chemotherapy, but shorter time pharmacoparameters were employed to study toxic side-effects. Several performance metrics, such as the AUC, the Harrell’s concordance C (C-index), the SE, the SP, the accuracy (ACC), the positive predictive value (PPV) and the negative predictive value (NPV), the classification precision and F1 scores (the harmonic means of the precision and recall) and patient survival estimation by hazard ratio (HR) were used to evaluate the predictive power of various algorithms for correctly classifying response vs. non-response.

### 3.2. Publication Validation

When data quality was assessed in the 26 AI study included in the systematic review and the meta-analysis [74,75,79,80,81,82,83,84,85,86,87,88,89,90,91,92,93,94,95,96,97,98,99,100,101,102], we identified a common gap that consisted in the lack of an external validation cohort, that was missing in more than the half of the investigations (Table 4 and Appendix A). Outliers, missing values and C.I. were also not handled by most of the studies.

### 3.3. Artificial Intelligence Predictive Models

In the following paragraphs, we provide a perspective of the progress in computer science by reviewing the roles that learning techniques have played in cancer research with predictive algorithms of response to chemotherapy alone or combined with targeted therapy as well as for predicting individual toxicity for mCRC patients. As shown in Table 2 and Table 3, 26 studies were conducted to develop and validate different predictive algorithms for mCRC patients [74,75,79,80,81,82,83,84,85,86,87,88,89,90,91,92,93,94,95,96,97,98,99,100,101,102]. In the AI investigations, cohorts were generally divided into a training and a validation set to train and validate the developed models to avoid overfitting, with the exception of three studies [79,92,93], that did not split the data. Data on training and validation set were also not always reported. In radiomics studies, different timepoints in the same person were often used in both the training and testing sets, e.g., at baseline and after chemotherapy. The distribution of the outcome cases, e.g., numbers of responders vs. non-responders, was commonly different in the testing and training data sets of the 26 AI investigations. Information on the distribution of the outcome cases, e.g., numbers of responders vs. non-responders, was generally missing, with the exception of three AI studies, where the following frequency were reported: 131 responder vs. 61 non-responders in the study of Wei et al. [80], 42 responder vs. 41 non responders in the study of Tian et al. [95], and 184 responders vs. 210 non responders in the study of Liu et al. [94].

### 3.4. Cytotoxic Chemotherapy and Radiomics Learning Models

As mentioned above, oxaliplatin is widely used as part of first-line therapy in mCRC patients. However, only some patients achieve OR, and most of them will progress in the subsequent months [5]. To evaluate the treatment response of mCRC to first-line FOLFOX chemotherapy, Giannini et al. [75] have employed several ML networks to build a radiomic signature using texture features from contrast-enhanced computed tomography (CT) scans of liver metastasis performed at baseline and at subsequent time points. In that investigation, the authors published a delta-radiomic signature capable of correctly identifying lesions as responder vs. non-responder to oxaliplatin-based regimen. The best performance was shown by a tree-structured algorithm in respect to other networks, a SVM and a random forest, with AUCs of 0.99 (95% C.I. 0.97–1.0, 99% SE, 94% SP, ACC of 97%) and 0.93, (95% C.I. 0.87–0.96, 85% SE, 92% SP, ACC of 86%) in training and validation sets, respectively. In another study, Nakanishi et al. [79] developed computer-aided prognostic models aiming to distinguish mCRC patients suitable for standard chemotherapy. Specifically, they used a learning regression model to construct a radiomic signature for assessing the response to oxaliplatin-based first-line chemotherapy using the texture features from pretreatment contrast-enhanced CT scans of liver metastases. The radiomic signature built was able to discriminate responders vs. non-responders and the performance of the learning model reached AUCs of 0.851, 95% C.I. 0.771–0.930 and 0.779, 95% C.I. 0.617–0.940, in training and validation arms, respectively. 

Using a computational approach, Wei et al. [80] have evaluated the power of a DL model to predict the response to first-line regimens including oxaliplatin or irinotecan for mCRC or recurrent CRC patients using data from pretreatment contrast-enhanced multidetector tomography (MDCT) scans of liver metastases. The radiomic signature constructed by a residual network architecture algorithm reached a high predictive performance for chemotherapy response with an AUC of 0.903, 95% C.I. 0.851–0.955. The classifier was found to be able to identify responders vs. non-responders with 84.7% SE, 84.8% SP, ACC of 84.7% on the training set, but lower estimates were observed in the validation set (AUC of 0.745, 95% C.I. 0.659–0.831, 90% SE, 73% SP and ACC of 85.4%). A better performance was instead obtained by the final fusion radiomic signature that integrated the response of carcinoembryonic antigen serum level, with AUCs of 0.935, 95% C.I. 0.897–0.973 and 0.830, 95% C.I. 0.688–0.973 on training and validation sets, respectively. 

To identify radiomics signatures predictive of response to standard chemotherapy (i.e., first-line FOLFOX or FOLFIRI) for mCRC patients, Defeudis et al. [81], have developed and validated several ML technologies for predicting the response to first-line FOLFOX or FOLFIRI using data derived from pretreatment CT scans of metastatic lesions. In that study, different radiomic features were fed as inputs to the learning model, including those extracted from the whole metastasis by a 3D approach and others from the traditional region of interest. The best performance in identifying responder vs. non-responder lesions was given by a SVM classifier with 3D data approach rather than by a Gaussian naive Bayes or a multilayer perceptron algorithms in *per*-*lesion* analysis. The performance of SVM showed 76% SE, 67% SP, ACC of 72%, 69% PPV, 75% NPV and 61% SE, 60% SP, ACC of 61%, 64% NPV, 57% PPV in training and validation sets, respectively. Conversely, when SVM algorithms were carried out in *per*-*patient* analysis, the model yielded only an ACC of 32%, 41% SE and 21% SP, with 23% NPV and 38% PPV on validation set. 

### 3.5. Cytotoxic Chemotherapy Alone and Biomarkers and Clinical Learning Models

Tian et al. [95] have investigated the response to FOLFOX chemotherapy by ML technique using publicly available genomic data of mCRC patients from GSE28702 and GSE72970 cohorts. In this study, a 74-gene signature, identified by different learning algorithms in the top rank 250 genes, was used by an iterative supervised learning (IML) method to classify mCRC cases into responder or non-responder. The learning prediction model showed a SE of 97.6% and a SP of 100% in the training set, whereas the survival analysis resulted in a HR of 2.6 in the validation set with a median OS time of 13.4 months in non-responders vs. 36.6 months in responders. The pattern of gene signature of non-responders showed increased expression of *ERCC excision repair 1, endonuclease non-catalytic subunit* (*ERCC1*) and *dihydropyrimidine dehydrogenase* (DPYD) genes, suggesting high catabolism rate and efficient nucleotide excision repair.

Long noncoding RNAs (lncRNAs) have been involved in most of fundamental tumor processes including epithelial-to mesenchymal transition (EMT), immunity and angiogenesis [103,104], through a lncRNA transcriptome analysis that identified lncRNAs associated with these and other features in CRC. However, the clinical significance of lncRNAs is still unclear. To explore this issue, Liu et al. [84] have employed different ML models to develop and validate an immune related signature able to predict the response to 5-fluorouracil-based regimens with or without anti-VEGF therapy. In particular, immune infiltration data of CRC patients, including cohorts of patients with CRC metastasis, from the TCGA Research Network portal (https://portal.gdc.cancer.gov/ (accessed on 30 April 2022)) and Gene Expression Omnibus (GEO, http://www.ncbi.nlm.nih.gov/geo/ (accessed on 30 April 2022)) were used. In this system, 235 lncRNA modulators of immune-related pathways, including the T cell receptor signaling and antigen processing and presentation pathways, were identified by integrated algorithms using molecular data derived from immune infiltration patterns of the TCGA-CRC database (https://portal.gdc.cancer.gov/ (accessed on 30 April 2022)). Forty-three potential predictive lncRNAs were then used to build a consensus immune-related lncRNA signature (IRLS) by an integrative combination of Lasso and stepwise Cox regression, which identified a final signature formed by 16 predictive lncRNAs. The classifier was formed by immune-related lncRNAs encoded from the following RNA genes, the *imprinted maternally expressed transcript* (*H19*), the *long intergenic non-protein coding RNA 308* (*LINC00308*), the epithelial cellular adhesion molecule (EpCAM), the *SEPTIN7 divergent transcript* (*SEPTIN7-DT*), the *ASH1 like histone lysine methyltransferase* (*ASH1L*), the *MRGPRG antisense RNA 1* (*MRGPRG-AS1*), the *MIR210 host gene* (*MIR210HG*), the *uncharacterized LOC100507250* (*LOC100507250*), the *fibrous sheath interacting protein 1* (*FSIP1*), the *leucine-rich repeat-containing protein 61* (*LRRC61-1*) and the *long intergenic non-protein coding RNA 2560* (*LINC02560*). In this study, responders to chemotherapy were characterized as having higher levels of IRLS signature than non-responders, whereas the levels of IRLS signature were decreased in metastatic or recurrent CRC patients who were sensitive to anti-VEGF therapy. In particular, the IRLS signature was found to predict the response to fluorouracil-based regimens for metastatic or recurrent CRC patients on the GSE19860 (AUC of 0.843), GSE69657 (AUC of 0.765), and GSE72970 (AUC of 0.709) cohorts, as well as the benefits of adding bevacizumab on the GSE19860 (AUC of 0.771), GSE72970 (AUC of 0.781) and GSE19862 (AUC of 0.694) cohorts. Of note, performance estimates might reflect small sample size of different GSE cohorts (Table 2 and Table 3).

Prognosis for mCRC can be influenced by different gene expression profiles [105]. Hence Lu et al. [101] have investigated the performance of several ML techniques to predict the response to first-line FOLFOX chemotherapy using gene expression data. In this investigation, the authors identified 18 differentially expressed genes (DEGs), that were enriched in autophagy, ErbB signaling pathway, mitophagy, endocytosis, FoxO signaling, apoptosis, and antifolate resistance pathways, between FOLFOX responders and non-responders. Based on the 18-gene classifier, a random forest was able to predict the response to FOLFOX chemotherapy with a performance (AUC) of 0.877, 95% C.I. 0.747–1.00, with 85% in SE and 69.2% in SP in validation set. In addition, the authors demonstrated that the over expression of the *mixed lineage kinase domain like pseudokinase* (*MLKL*) and the *coiled-coil domain containing 124* (*CCDC124*) genes were associated with increased survival (HR of 0.358, 95% C.I. 0.178–0.717 and 0.563, 95% C.I. 0.336–0.943, respectively). However, when the learning models were applied to predict the response to FOLFIRI chemotherapy, the best model was the neural network algorithm with an AUC of 0.778, 95% C.I. 0.576–0.979. 

Chen et al. [83] had the aim of constructing a learning model predictor for the response to oxaliplatin- or irinotecan-based regimens using circulating cytokines in mCRC patients. In that study, the authors identified a predictive signature of 17-cytokines using univariate SVMs, including the fibroblast growth factor-2 (FGF-2), a growth factor and signaling protein, the transforming growth factor α (TGFα), an angiogenic *cytokine and* an endothelial growth factor, the *cytokine* Fms-like tyrosine kinase 3 ligand (Flt3), an important regulator of hematopoiesis, the human interferon α-2 (INFα2), the inflammatory interleukins 2, 7, 8 and 10 (IL-2, IL-7, IL-8, and IL-10), the soluble form of IL-*2* receptor (sIL-2Ra), the *tumor necrosis factor* α (TNFα), the inflammatory cytokines monocyte chemotactic proteins 1 and 3 (MCP-1 and MCP-3), the VEGF, the macrophage-derived chemokine (MDC), and the granulocyte-macrophage colony-stimulating factor (GM-CSF) [106]. Based on the 17-cytokine signature, a COX proportional hazard model was able to predict a worst survival in the group of high-risk patients characterized by high-cytokine expression, with an overall SE of 83.5%, 80% in SP, and an ACC of 81% on the training set. Comparable estimates were computed on the validation set (83.1% in SE, 66.7% in SP, and 74.9% ACC). Moreover, high-risk patients showed a worst OS as compared to low-risk patients in both training and validation arms.

Tsuji et al. [82] utilized a random forest to build a learning model that could predict the response to first-line FOLFOX chemotherapy using gene-expression data from mCRC patients. In this study, the authors determined a gene signature for FOLFOX based on the gene expression of 14 classifier genes. In more detail, the 14-gene signature was constituted by different genes, including the *smad ubiquitin regulatory factor 2* (*SMURF2*), a negative regulator of TGF-β signaling with a tumor-suppressive role [107], the *Mbt domain containing 1* (*MBTD1*), encoding a potent transcriptional coactivator [108], the *adaptor related protein complex 3 subunit Mu 2* (*AP3M2*), the *RING finger protein 141* (*RNF141*), a gene reported to interact with *KRAS* promoting CRC progression [109], the *aminopeptidase puromycin sensitive* (*NPEPPS*), encoding key zinc metallopeptidases [110], the *bromodomain PHD finger transcription factor* (*BPTF*), a chromatin remodeling-related gene [111], the *family with sequence similarity 73, member A* (*FAM73A*), the *amyloid beta precursor protein binding protein 2* (*APPBP2*), the *archaelysin family metallopeptidase 2*
*pseudogene* 1 (*AMZ2P1*), the *SLIT-ROBO rho GTPase activating*
*protein*
*1* (*SRGAP1*), that encodes a protein mediating cell migration associated with CRC tumor progression and poor prognosis [112], the *N-myristoyltransferase-1* (*NMT1*), necessary for lysosomal degradation and mammalian target of rapamycin (mTOR) signaling pathway [113], the *centrosome and spindle pole associated*
*protein*
*1* (*CSPP1*), whose circ-CSPP1 was found to be significantly overexpressed in CRC tissues [114], the *eukaryotic translation initiation factor 1* (*EIF1*), and the *centrosomal protein 290* (*CEP290*) genes. Using the 14-*gene* classifier, the learning model was capable of classifying responders with an overall SE of 91.3% compared with non-responders with 95.6% in SP, achieving an ACC of 80.2%. In addition, responders showed an extent of 22.7 more months of median OS compared with non-responders in the training set.

Yuan et al. [84] have constructed a protein-based prediction model for analyzing the response to first-line with FOLFOX or FOLFIRI regimens for mCRC patients. Using this protein-based algorithm, responders to both treatments were identified with an overall SE of 92.9% and an 81.3% SP for FOLFOX and a SE of 92.3% and 92.3% SP for FOLFIRI regimens. Similarly, Del Rio et al. [92] have used ML approaches to build a predictor of chemotherapy response using gene expression data for mCRC receiving FOLFIRI regimens. By a learning approach, a pattern of 14 highly discriminatory genes involved in several pathways was investigated for assessing the potential predictivity. The main pathways were including RNA splicing (*U2AF1L2*), regulation of transcription (*ZNF32* and *ZNF582*), cell adhesion (*F8*, galectin-8, *PSG9*), cell differentiation (*SERPINE2*, *BOLL*), ion transport (*ATP5O*), signal transduction (*DRD5*), development (*ANGPTL2*) and visual perception (*EML2*). In this investigation, a SVM model reached 92% SE, 100% SP and 95% ACC in classifying mCRC patients and identifying the subset of them who could benefit from FOLFIRI chemotherapy.

Irinotecan-based regimens can cause a heavy collateral effect leading to the delay or even the suspension of chemotherapy with a subsequent impact on the survival of mCRC patients. The study of pharmacokinetic parameters is a valuable option to improve both efficacy and safety of cancer drugs administration. In mCRC treatment, pharmacokinetic studies have revealed that systemic exposure to irinotecan is highly variable among patients. Therapeutic drug monitoring and genotype-driven studies might thus be helpful in applying dosage adjustments for this drug. Oyaga-Iriarte et al. [93] developed various ML algorithms for predicting side effects of irinotecan regimens using pharmacokinetic parameters in mCRC patients. In this network, the best mathematical model for predicting leukopenia in patients with liver metastasis was built by random forests (AUC of 0.74, 89% in SE, 60% in SP, and 76% ACC). The more performant model for predicting neutropenia was the supervised SVM algorithm (AUC of 0.88, 70% in SE, 70% in SP, and 75% ACC), while the best learning model for predicting diarrhea was the backward stepwise logistic regression (AUC of 0.95, 81% in SE, 100% in SP, and 91% ACC). 

To develop a predictive model for the risk of cardiotoxicity, a severe side effect of standard chemotherapy, Li et al. [96] employed different ML methods to analyze the outcome of 30-day cardiotoxicity—using any cardiotoxicity event—on a large cohort of 36,030 CRC patients undergoing fluoropyrimidine-based chemotherapy. The cardiotoxicity events were correlated to six broad classes of variables available in the SEER-Medicare database, including anticancer treatment information, cancer information, demographic variables, socioeconomic status, prior cardiovascular disease histories and other comorbidities, prior medications. The algorithm reaching the highest performance was an extreme gradient boosting (XGBoost) algorithm, based on cardiovascular features such as ischemic heart diseases, cardiomyopathy, arrhythmia and stroke, that showed a precision of 0.621, F1 score of 0.396 and an AUC of 0.801, 95% C.I. 0.781–0.821 in predicting cardiotoxicity in mCRC patients.

### 3.6. Targeted Therapy and Radiomics Learning Models

Tumor response assessment is commonly based on tumor size change on instrumental images, but morphological changes in tumor can occur earlier in response to antiangiogenic agents. To face this limit, Lu et al. [85] utilized data derived from pretreatment and follow-up CT scans of metastases and lymph nodes to construct DL networks aimed to predict the response to anti-VGEF agents in mCRC patients from the VELOUR trial [115]. In this investigation, the DL network architecture constructed with CNN and RNN networks and based on the quantitative characterization of tumor morphological changes, was able to give an early prediction of treatment response. The performance of the DL model showed C-indexes of 0.678, 95% 0.650–0.706 and 0.649, 95% C.I. 0.619–0.679 on training and validation arms. The integration of the DL network with traditional size-based methodology still improved the performance with a C-Index of 0.694, 95% C.I. 0.661–0.720. The DL network also reached higher power in classifying responders and non-responders to antiangiogenic drugs with responders having a better OS than the non-responders, with median OS 18 vs. 10.4 months, respectively (HR of 0.49, 95% C.I. 0.40–0.61).

Learning models combined with the analysis of *RAS* mutations can offer important advantages for predicting the response to treatments of CRC metastases. Patients with tumors carrying *RAS* mutations are in fact considered non-responders to anti-EGFR therapy whereas those with wild-type *RAS* status are not always sensitive to anti-EGFR antibodies due to the less frequent mutations in the EGFR signaling pathway [39]. To correlate radiomic parameters to *RAS* mutational status, Granata et al. [91] have examined the association of *RAS* mutational status with anti-EGFR therapy by ML using data derived from pretreatment contrast enhanced magnetic resonance imaging (MRI) of liver metastases in mCRC patients from the OBELICS cohort [116]. Using a radiomic signature based on both textures and morphological features, *RAS* mutational status was better detected by a supervised SVM algorithm as compared to other learning algorithms, as a k-nearest neighbors (KNN), an ANN, and a DT network (AUCs of 0.79, 95% C.I. 0.70–0.85, 78% SE, 74.2% SP, ACC of 76.1% and 83.3% SE, 75% SP, ACC of 79.2% in training and validation sets). However, the best result was obtained by a supervised KNN algorithm exclusively fed with robust textures as predictors (AUC of 0.84, 95% C.I. 0.780–0.91, 90% SE, 67.8% SP, ACC of 76.9% and 91.7% SE, 83.3% SP, ACC of 87.5% on training and validation arms).

Zhu et al. [98] analyzed the response to anti-VEGF therapy by DL, using features extracted from preoperative and post-operative MRI of no more than five liver metastatic lesions in mCRC patients. Using a densely connected center cropping CNN (DC3CNN) architecture, features from input data, pre-treatment T2-weighted image, post-treatment data, were extracted to construct three predictive models. The best performance for differentiating non-responder and responder patients was obtained by the DL model containing all the features with an AUC of 0.849, 95% C.I. 0.737–0.926, 91.7% SE, 75% SP, ACC of 87.5%, 75% NPV, 91.7% PPV in the validation set. In addition, the performance of the radiomic signature was further validated in an external independent cohort, with AUC of 0.833, 95% C.I. 0.695–1.000, 91.9% SE, 75% SP, ACC of 88.5%, 69.2% NPV, 93.8% PPV. DL also demonstrated better PFS and OS in responder vs. non-responder groups.

Maaref et al. [86] utilized a DL computational tool to predict the response to anti-VEGF therapy using the textures derived from pretreatment contrast enhanced CT scans of liver metastases of mCRC patients. In this work, a fully automated network (DCNN) achieved higher performance to identify new lesions appearing on CT images by classifying tumors in “treated” vs. “untreated lesions” as compared to other algorithms, as DT, SVM, and ANN models, with AUCs of 0.97 vs. 0.66, 0.60 and 0.62, respectively. The unsupervised DCNN algorithm showed early detection of non-responsive patients with respect to all untreated lesions with AUCs of 0.83 (95% C.I. 0.78–0.87, 97% SE, 59% SP, ACC of 78%) and 0.88 (95% C.I. 0.85–0.94, 98% SE‚ 54% SP, ACC of 76%) in the training and validation arms, respectively. 

Dercle et al. [87] conducted a study aiming to develop an algorithm predictive of tumor sensitiveness to standard chemotherapy with or without anti-EGFR agents by ML, using data derived from pretreatment and follow-up CT scans for mCRC patients from the CRYSTAL cohort [9]. The authors observed that the radiomic signature based on spatial and temporal heterogeneity of tumors was highly performant to predict treatment-sensitive tumors to anti-EGFR targeted therapy. The best predictive model was constructed by a random forest algorithm using high quality medical images with AUCs of 0.83 (95% C.I. 0.75–0.92, 77% SE, 85% SP) and 0.80 (95% C.I. 0.69–0.94, 80% SE, 78% SP) in the training and validation sets, respectively. Comparable results were obtained using standard of care scans with AUCs of 0.84 (95% C.I. 0.76–0.89) and 0.72 (95% C.I. 0.59–0.83) in the training and validation arms, respectively. The high-risk radiomic signature was associated to shorter OS as compared to the patients with low-risk signature in both training and validation sets. However, such radiomic signatures failed to predict sensitivity when the response to chemotherapy alone was investigated. 

Vera-Yunca et al. [88] constructed a predictive model of response to anti-EGFR therapy by learning algorithms using features derived from pretreatment CT scans and MRI for mCRC patients from the CRYSTAL [9], APEC [117], Study 045 [118] and OPUS [119] cohorts. In this investigation, the k-means clustering constructed with data of tumor heterogeneity in lesion dynamics was predictive of OS in mCRC patients. The model showed that high tumor heterogeneity was associated with a worst OS with an HR of 1.44, 95% C.I. 1.08–1.92, mostly in patients with *KRAS* mutated tumors.

Recently, anti-HER2 therapy has proven to be a beneficial treatment option for mCRC patients with HER2 amplification or overexpression [43]. Therefore, to predict the response to dual anti-HER2 therapy, Giannini et al. [74] developed and validated ML networks predictive of the response to trastuzumab and lapatinib using data from baseline CT scans of liver metastases of mCRC patients with *RAS* wild-type and HER2 amplified tumors from the HERACLES cohort [120]. In this network, a Gaussian naïve Bayesian classifier reached high performance in identifying responder vs. non-responder lesions in patients with heterogeneous response, with 89% SE, 85% SP, 93% NPV, 78% PPV, and 90% in SE, 42% in SP, 73% NPV, 71% PPV, in training and validation lesions sets, respectively. Comparable high predictive values were also obtained by the radiomic score in predicting overall patient response with 92% SE, 86% SP, 75% NPV, 96% PPV in patient arm.

### 3.7. Targeted Therapy and Biomarkers Learning Models

The reported limited clinical efficacy of angiogenesis inhibitors [5] has prompted the need for better understanding of resistance mechanisms and the elucidation of novel predictive models. To direct attention to this item, Abraham et al. [99] studied the response to anti-VEGF therapy, including bevacizumab plus FOLFOX or FOLFOXIRI regimens, by a learning approach using data from next generation sequencing (NGS) in mCRC patients from the TRIBE2 cohort [121]. In that investigation, the genomic features of 592-gene data fed as inputs into more than one hundred learning models, produced a final model containing a 64-gene classifier. This 64-gene predictive classifier, named FOLFOXai, consisted of several genes involved in relevant pathways. The following genes were included in the signatures: the mediating WNT signaling (*BCL9* and *CDX2*), epithelial-to-mesenchymal transition (*EMT*, *INHBA*, *PRRX1*, *PBX1*, and *YWHAE*), chromatin remodeling (*EP300*, *ARID1A*, *SMARC4*, and *NSD3*), DNA repair (*WRN* and *BRIP1*), NOTCH signaling (*MAML2*), and cell-cycle regulation (*CNTRL* and *CCNE1*). By FOLFOXai, mCRC patients were distinguished in “increased benefit” (IB) vs. “decreased benefit” (DB), where the category of IB showed improved clinical outputs in both oxaliplatin-containing arms. Moreover, the signatures showed a median TTNT of 11.5 months for IB and 8.2 months for DB, HR of 0.537, 95% C.I. 0.428–0.674, as well as a median OS of 42 months for IB and 24.5 months for DB, HR of 0.466, 95% C.I. 0.325–0.670 in the FOLFOX/bevacizumab arm. Conversely, the FOLFOXai prediction resulted inverted in FOLFIRI cohort with a median OS of 18.7 months for IB and 34.4 months for DB, HR of 2.631, 95% C.I. 1.041–6.649. A median PFS of 9.6 months for IB vs. 8.7 months for DB, HR of 0.757, 95% C.I. 0.505–1.135 and a median OS of 24.8 months for IB vs. 18.7 months, HR of 0.629; 95% C.I. 0.404–0.981 were detected in the FOLFOX/bevacizumab arm. Improved median PFS of 13.8 vs. 7.6 months, HR of 0.683, C.I. 0.396–1.181 and higher median OS of 30 vs. 15.9 months, HR of 0.483, C.I. 0.270–0.864 were even reported in the FOLFOXIRI/bevacizumab arm. 

*KRAS* mutation profiles status can predict the response to treatment in mCRC [39], therefore Naseem et al. [102] have investigate by random survival forests method selected single nucleotide polymorphisms (SNPs) potentially predictive of the response to anti-VEGF therapy, including bevacizumab plus FOLFIRI, in both *KRAS* mutant and wild-type mCRC patients from the TRIBE and FIRE-3 cohorts [121]. Among 27 SNPs, the learning algorithms identified that SNPs in Wnt/β-catenin, tumor associated macrophage and sex-differentiation pathways were predictive of OS and/or PFS in mCRC patients treated with FOLFIRI/bevacizumab. In detail, the *CBP* rs129963 T/T variant and the β-catenin rs3864004 A/A genotype were associated with worst OS and PFS in *KRAS* wild-type (OS of 22.8 vs. 26 months and PFS of 9.5 vs. 10.5 months) and mutant (OS of 16.3 vs. 26.3 months and PFS of 7.8 vs. 9.6 months) patients, respectively. Conversely, improved OS and PFS were observed in both *KRAS* mutant (PFS of 10.3 vs. 8.6 months) and wild-type (OS of 31.3 vs. 24.8 months and PFS of 11.3 vs. 10.3 months) patients with the TBK1 rs7486100 A/A variant. Higher estimates were also obtained for the *CCL2* rs4586 T/T carriers (OS of 30.9 vs. 22.8 months), the *VEGFR2* rs2305948 any C carriers (OS of 26.2 vs. 17.0 months) and in the *MMP2* rs243865 any T carriers (OS of 28.5 vs. 20.3 months). 

In order to identify single nucleotide polymorphisms (SNPs) potentially predictive of the mechanisms of response and resistance associated to anti-VEGF therapy, Barat et al. [89] analyzed by learning methods the association between SNPs and the response to bevacizumab plus FOLFIRI chemotherapy using data from exome-sequencing in mCRC patients from the ANGIOPREDICT, MAVERICC and TRIBE cohorts [121]. In this study, 47 novel predictive SNPs were selected by learning algorithms across all exomes. Along the novel SNPs, two SNPs alone, the *NLRP1* rs12150220 and the *SRL* rs13334970, one of which encodes a pro-inflammatory gene (*NLRP1*), were found to be associated with response to anti-VEGF therapy. For the rs12150220, missense at amino acid 155 of *NLRP1* (*NLR Family Pyrin Domain Containing 1*), TT carriers had better median PFS than AA carriers in the ANGIOPREDICT and MAVERICC cohorts, HRs of 0.52, 95% C.I. 0.33–0.83 and 0.42, 95% C.I. 0.21–0.85, respectively. Conversely, for the rs13334970, mapping to *SRL* (*Sarcalumenin*), AA carriers showed shorter median PFS than carriers of at least one G allele in ANGIOPREDICT and MAVERICC cohorts, HRs of 2.3, 95% C.I. 1.19–4.57 and 2.5, 95% C.I. 1.12–5.5, respectively. The *NLRP1* any A and the *SRL* AA carriers had worse PFS in ANGIOPREDICT and MAVERICC datasets, HR of 8.3, 95% C.I. 3.3–21 and 2.2, 95% C.I. 1–5, respectively. Association with worse outcomes was found with *KRAS* wild-type and concomitant carriers in combination with the *NLRP1* any A and the *SRL* AA, HRs of 7.8, 95% C.I. 2.5–24.4 and 3, 95% C.I. 1.2–8 in MAVERICC and TRIBE. In *KRAS* wild-type patients, the *NLRP1* any A and the *SRL* AA carriers showed a worse PFS than in the other two genotype combinations with shorter median PFS (4 vs. 11 and 15 months), and also shorter median OS (12 vs. 27 months).

Williams et al. [97] have examined by a ML approach using immunohistochemical data the response to anti-EGFR therapy with or without irinotecan regimens in mCRC patients with *KRAS* wild-type from the PICCOLO dataset [122]. By whole slide learning and computer vision techniques, the percentage of positively stained tumor cells within the tumor areas for EGFR ligands amphiregulin (AREG) and epiregulin (EREG) were found to predict benefit from the anti-EGFR agent panitumumab. Higher PFS was associated to the response to targeted therapy in patients with *KRAS* wild-type and high ligand expression as compared to those receiving standard regimens (8.0 vs. 3.2 months, HR of 0.54, 95% C.I. 0.37–0.79), and in those with both *KRAS* and *BRAF* wild-type, HR of 0.53 (0.36–0.78).

Ubels et al. [90] have recently evaluated whether genome-wide genotyping could be used to predict the efficacy of cetuximab in mCRC patients from the CAIRO2 dataset [123]. In this study, a random forest approach based on 781-SNP signature was able to classify mCRC patients in two classes, “benefit” vs. “no benefit”; a subset of patients showed benefit from receiving anti-EGFR therapy demonstrated by a higher median PFS (HR of 0.69, 95% C.I. 0.49–0.98), as compared to the category ‘no benefit’ (HR of 1.32, 95% C.I. 1.07–1.62). A greater performance was even found when chromosomal sex was incorporated in learning models (HR of 0.52, 95% C.I. 0.35–0.76).

Johnson et al. [100] have investigated by a ML approach with gene-mutation data the response to standard chemotherapy with or without target therapy in mCRC patients from the MSK cohort [124]. In this study, mutation profiles fed as inputs into a RF model, produced a final model containing a classifier based on mutation profiles of *KRAS*, *BRAF*, *ERBB2*, *MAP2K1*, *TSC2*, *TP53*, and *APC*. The 7-gene algorithm showed high predictive power for PFS to identify responder (non-progressed) vs. non-responder (progressed) mCRC patients with HR of 16.9, 95% C.I. 4.2–68.0.

### 3.8. Meta-Analysis

We performed a meta-analysis of the 9 AI studies reporting performance estimates of the response to therapy and 95% C.I. with 2.441 patients [75,79,80,85,86,87,98,99,101]. Table 5 reports the estimates of AUCs and HRs and their C.I. used for discriminating response to therapy that were extracted from the predictive models from the aforementioned studies. Results of meta-analysis are shown in Figure 2 reporting the single study and the overall weighted mean of AUC and HR with their 95% C.I. The sensitivity analysis showed that no study heavily influenced the result of this meta-analysis. In addition, it was found that the occurrence of publication bias was unlikely. Using the random-effect model, we calculated summary risk estimates in training and validation sets. Overall weighted means of AUCs are 0.90, 95% C.I. 0.80–0.95 and 0.83, 95% C.I. 0.74–0.89 in training and validation sets, respectively, showing good classification performance in discriminating response vs. non-response. On the other hand, an overall HR of OS of 0.51, 95% C.I. 0.43–0.60 was found for the learning models that significantly predict improved survival (Figure 1).

Using the random-effect model, we calculated summary risk estimates in training and validation sets. Overall weighted means of AUCs were 0.87, 95% C.I. 0.80–0.92 and 0.79, 95% C.I. 0.69–0.87 in training and validation cohorts, respectively, showing high classification performance in discriminating responder mCRC patients/treated lesions vs. non-responder mCRC patients/untreated lesions to standard chemotherapy alone or combined with targeted therapy. On the other hand, overall HR of OS of 0.51, 95% C.I. 0.43–0.60 was found for learning models that significantly predict improved survival (Figure 1).

### 3.9. Predictive Model Performances

Model performances were then analyzed in 13 AI studies that reported AUCs, SE and SP estimates [74,75,79,80,81,86,87,94,95,98,99,101]. Considering the AI studies that have analyzed the model performance using the AUCs, Table 6 shows that both radiomics and molecular signatures were effective in predicting the response to therapy with good or excellent estimates in both the training and the validation sets. Comparable estimates of the global AUCs were also reported by our meta-analysis. Moreover, when we evaluated the AI investigation with tests that reached the 80% SE and the 90% SP, there were the delta radiomics that showed the SE of 99% and the SP of 94% in the training set and the SE of 85% and the SP of 92 in the test set, and the 74 gene signatures that obtained the SE of 97.6% and the SP of 100% in the training set.

## 4. Discussion

Selection of the optimal first-line treatment represents a crucial step in the therapeutic pathway of mCRC patients, in order to obtain a significant improvement of PFS and, possibly, OS due to the development and combination of cytotoxic and biologic drugs chosen on the basis of the tumor mutational status. However, biological drugs, being directed towards specific “actionable” targets, might cause a heterogeneous tumor response, depending on clinical characteristics and/or disease biology of each patient. Therefore, despite selection of optimal therapy based on the patient’s molecular phenotype, a percentage of patients is not responsive to targeted therapy, leaving us to hypothesize that additional mediators could be involved in the dysregulation of molecular mechanisms, that expression of further potentially actionable genes might intervene and/or that mechanisms of innate/acquired resistance to target inhibitors occur. Even the presence of genetic mutations in tumor *RAS* or *BRAF* sequences and MSI status cannot always predict the therapeutic response in mCRC patients [5]. No imaging criteria are also available that could predict the response to therapy before the start of the therapy and or even predict the kind of metastatic lesions that will respond to the treatment [125]. To face these limits, the design of optimal strategy for mCRC on a case by case basis has been proposed, where therapeutic interventions should be modulated depending on patient/tumor’s characteristics. Therefore, it is relevant to characterize CRC complexity at the individual level to understand if a patient will respond to a specific therapy or will show resistance, considering the opportunity to leverage new emerging computer science solutions from research to real-world conditions. In this regard, growing emphasis has been put on clinical decision support systems based on AI, in general, and ML techniques, in particular, to develop predictive models of cancer progression or response to treatment using ML techniques, especially in mCRC. This approach has already proven capable of exploiting significant patterns in routinely collected demographic, clinical and biochemical data and allowed the design of clinical decision support systems (DSS) that can be easily adapted to different tumors [126,127].

Our review summarizes the current literature exploring the use of machine and deep learning methods in predicting the response to cytotoxic and/or targeted treatments in mCRC patients. Table 1 and Table 2 show that most of the studies have benefited from the availability of high-throughput information from radiomics data and different genomic characterization levels, as genome, transcriptome, proteome, and metabolome. All investigations conducted in this field have shown promising outcomes. As reported in Table 2, ML networks were mostly utilized to analyze standard therapy response in mCRC patients, with the exception of the study of Wei et al. [80] that employed a residual network architecture DL model. In five studies [75,79,80,81,87], imaging biomarkers, including delta and fusion radiomic signatures, were used to evaluate the response to therapy. Molecular biomarkers classifiers, ranging from lncRNAs to gene-expression signatures were chosen by the authors of the other ten investigations for developing predictive models that could assist clinicians in decision making processes [82,83,84,92,93,94,95,96,99,101]. Results in Table 2 show that the algorithm most predictive of response to standard chemotherapy in mCRC patients was built using a delta-radiomic signature [75], where a trees structured algorithm reached the highest performance by classifying patients into responders vs. non-responders (AUC = 0.93). Importantly, this ML signature evaluating temporal changes in tumor phenotype was created as intermediate surrogate biomarker to support treatment continuation decisions in clinical yardsticks. Inconclusive findings were instead reported by the investigation of Dercle et al. [87], where a temporal radiomic signature failed to predict the treatment response. Great predictive performances in both training and validation sets were obtained by a DL model based on a fusion radiomic signature [80]. 

On the other hand, when we have examined the findings of the eleven studies based on molecular biomarkers [82,83,84,92,93,94,95,96,99,101,102], we found that microarray gene expression data were more often chosen by the different authors for building molecular biomarkers based classifiers [82,92,95,101]. Noteworthy, molecular information from immune infiltration, circulating cytokine expression, and pharmacokinetics and cardiovascular parameters were also often employed to fed as inputs learning models aimed to classify mCRC patients in responders vs. non-responders, as well as for evaluating side effects of standard regimens [82,83,84,93,94,96]. Along the molecular biomarker-based learning models, the greatest predictive performances were obtained by both 16-long noncoding RNA (AUC = 0.843) and 18-gene signatures (AUC = 0.877) [94,101]. Even a multiple linear regression model was able to correctly evaluate the response to chemotherapy [83]. However, the prediction of FOLFOXai classifier resulted inconclusively in the FOLFIRI cohort [99]. In addition, results indicate that pharmacokinetics and cardiovascular features could be used to predict different side effects, including cardiotoxicity and leukopenia, caused by standard cytotoxic regimens [93,96].

An improved knowledge of the mechanisms involved in colorectal carcinogenesis and tumor progression has allowed the development of biological agents with targeted actions, as inhibitors of angiogenesis, EGFR-targeted therapy, and immunotherapy [5]. As shown in Table 3, ML models [74,87,88,89,90,91,94,97,99,100,102] were mainly utilized to analyze the response to targeted therapy rather than DL networks [85,86,98]. Findings show that the prognostic and predictive values of radiomics and molecular biomarkers in predicting the benefit of targeted therapy in CRC metastases have been extensively analyzed, with seven radiomic studies demonstrating the utility of diagnostic imaging in predicting clinical outcomes [74,85,86,87,88,91,98] and with seven molecular biomarkers based investigations that developed successful predictive molecular classifiers [89,90,94,97,99,100,102]. Other inputs can come from a recent study of Lindner et al. [128], where several signaling proteins and phosphoproteins associated with EGFR and other relevant cancer signaling pathways when examined across mouse models. In that investigation an ML approach allowed identification of a 14-phospoprotein signature associated with EGFR and other relevant cancer signaling pathways including PDK1, caspase-8, Shc, Stat3, p27, GSK-3β, HER3, PKC-α, EGFR, Akt, S6 ribosomal protein, HER3, NF-κB-p65 and Gab-1, potentially able to discriminate tumors sensitive and insensitive to targeted therapy.

In this context, our meta-analysis demonstrated a good performance power of the learning models used to predict response to chemotherapy alone or combined with targeted therapy by discriminating response vs. non-response. The calculation of overall HR also shows that learning models have strong ability to predict improved survival. Next, when we analyzed performance, both radiomics and molecular signatures showed the capacity to predict the response to therapy with good or excellent values. In particular, we found that both the delta radiomics and the 74 gene signatures were able to discriminate response vs. non-response by correctly classifying up to 99% of mCRC patients as responders and up to 100% of patients as non-responders. 

There are several limitations to this study. First, in some cases, data of the same cohort were shared between training and test sets, whereas the use of an external validation cohort, that is critical to ensure model accuracy, was noted only in the half of the publications. In addition, three studies did not split the data [79,92,93]. This can be due to the lack of appropriate cohorts or the lack of awareness of the importance of an external validation cohort before its application in clinical settings. Missing data and outliers were also not appropriately managed by most of the AI studies. Some of them also have small sample sizes that could impact on study outcomes and decrease the statistical power. Then, the size of the validation sets was generally lower than that observed in the training sets, possibly reflecting the low number of mCRC patients that were available overall. Nevertheless, the use of k-fold validation that allows proper stratification of predictive outcomes [94,101] has been performed by the majority of the studies (77%), even at greater rate than in a previous review [72]. 

AI techniques can provide novel methods for clinicians focusing on predictive modeling for conventional chemotherapy alone or combined with target therapy as compared to previous statistical methods, that were more based on the study of data associations [24,129,130]. AI is indeed becoming popular for developing predictive modeling and defining treatment effects in epidemiology. Improving these approaches in molecular epidemiology will be useful for explaining cancer risk factors, including nature and nurture factors [24,131,132,133,134], in public health. Until now, researchers have benefited from a variety of AI methods for supervised and unsupervised pattern analysis in large big data originating from multiple sources, including serial imaging and molecular biomarkers. The major challenges are data variability, inter- and intra-study, and the lack of large amounts of high-quality labeled data, which both limit the potentiality of ML and DL models [135]. Privacy related issues, challenges in the patient enrollment process and cumbersome bureaucratic procedures do not allow researchers to easily obtain high quantities and/or high quality data.

When resistance to treatments occurs, few treatment options are available for resistant mCRC that may be difficult to tackle through non-cross-resistant anticancer drugs for second and subsequent lines of therapy despite new diagnostic and therapeutic methods [10]. Currently, about 20–25% of mCRC patients with hepatic metastasis are resectable, 60–70% of distal mCRC patients will develop local or distal recurrence, while only 20% will achieve long-term remission [136]. Up to 41% of mCRC patients receiving routine care have a poor performance status, and have more toxic side-effects if they are treated as well as inferior survival outcomes [137]. Furthermore, we cannot stratify mCRC patients or those who are at risk of developing colorectal metastasis at diagnosis using biomarkers or other currently available techniques. Even the presence of genetic mutations in tumor *RAS* or *BRAF* sequences and MSI status cannot always predict the therapeutic response in mCRC patients [5]. On the other hand, AI signatures can reach better performance in distinguishing responders from non-responders in respect to routine clinical indicators, such as cancer stage, adjuvant therapies, surgery on primary tumor and RECIST criteria [5,80,83,85,98,100,138]. New classifications of mCRC based on AI parameters will soon emerge and we will be able to identify new subtypes of mCRC patients, for whom the definition will be based on combinations of radiomics and/or molecular signatures contributing to improve the clinical management of mCRC and clinical decision making. AI approaches might provide innovative tools to distinguish between responders and non-responders before treatment starts, to know if a mCRC patient will respond to a specific therapy or will show resistance. 

In the future, especially with regard to clinical trials, our findings suggest that mCRC patients would be randomized to treatment based on a predictive model vs. usual treatment. However, the possibility of employing predictive models in choosing the most effective and cost-effective therapeutic intervention for cancer patients relies on the development of improved and innovative AI based computational frameworks. These models require the evaluation of a huge amount of clinical data, with emphasis on factors influencing positive and negative responses, prognostic biomarkers, and molecular predictors of therapeutic response or disease to be appropriately advanced to make such trials ethically correct. A rigorous process must always underlie any AI model to ensure reliable prediction of mCRC patient response to treatments. In this field, we are awaiting the first results of the European Union funded study titled “Targeted therapy for advanced colorectal cancer patients, REVERT” (https://www.revert-project.eu/ (accessed on 30 April 2022)) aimed at strengthening this ambitious goal by addressing the specific challenge of understanding at system level the pathophysiology of mCRC cancer in patients responding well or poorly to therapies, to design optimal strategies for mCRC on a case by case basis, with therapeutic interventions modulated depending on the patient’s features. 

## 5. Conclusions

Our systematic review shows that a good performance in predicting the response to therapy was obtained by the different algorithms that were analyzed by a meta-analytic approach. The calculation of overall HR indicates that learning models have a strong ability to predict improved survival. In addition, the delta-radiomics and the 74 gene signatures were found to able to discriminate between response vs. non-response by correctly identifying up to 99% of mCRC patients who are responders and up to 100% of patients who are non-responders. Our findings support the use of computer science for developing personalized treatment decision processes for mCRC patients. The identification of clinical, pathological and molecular comprehensive markers/signatures predictive of efficacy and toxicity could be useful for validation in prospective clinical trials. Further understanding of CRC biology linked to AI approaches will be able to improve matching mCRC patient individual traits with appropriate therapies to increase their survival and quality of life.

## Figures and Tables

**Figure 1 cancers-14-04012-f001:**
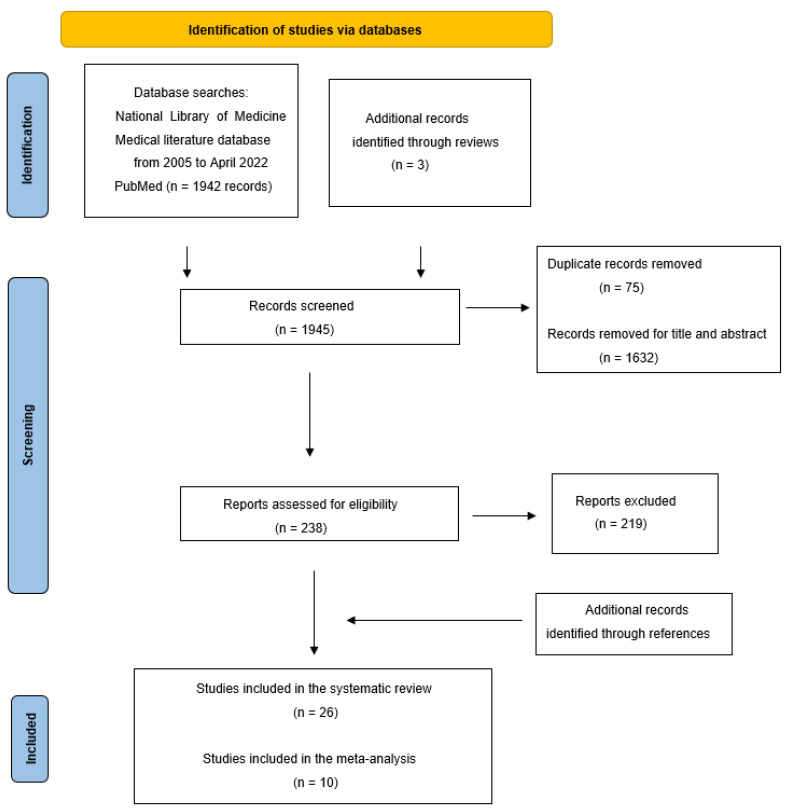
Systematic review flow diagram. The PRISMA flow diagram for the systematic review detailing the database searches, the number of abstracts screened and the full texts retrieved.

**Figure 2 cancers-14-04012-f002:**
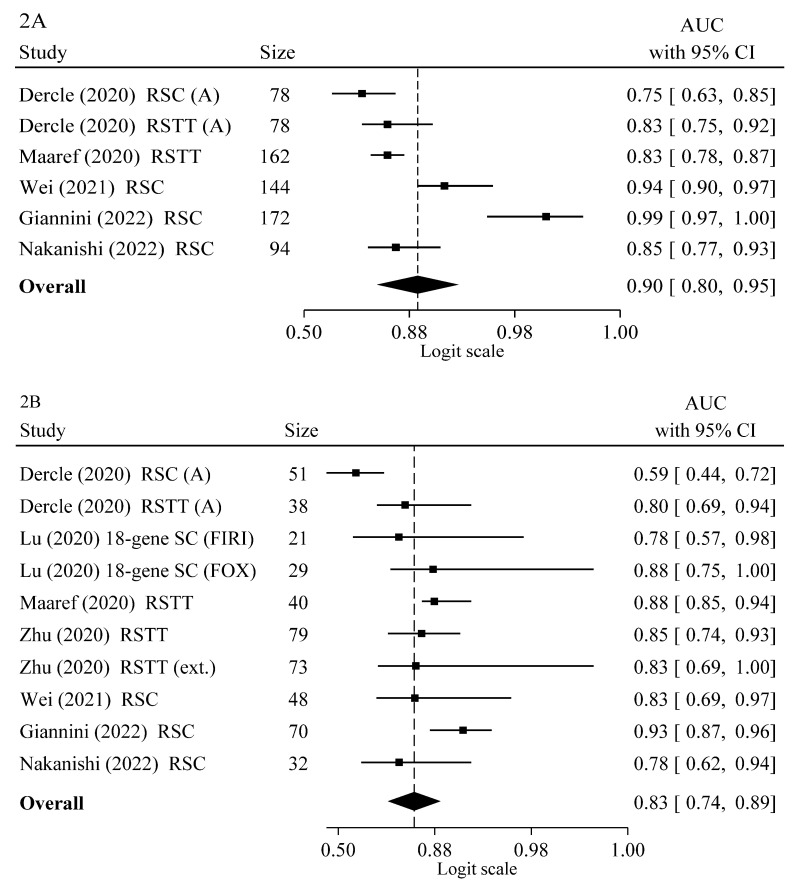
Meta-analysis of 21 artificial intelligence predictive models of response to therapy for metastatic colorectal cancer (mCRC) patients. The efficacy of learning algorithms in discriminating responders mCRC patients/treated metastatic lesions vs. non-responders mCRC patients/untreated metastatic lesions was analyzed by estimating the area under the receiver operating characteristic (ROC) curve (AUC) and 95% Confidence Intervals (C.I.) in training (**A**) and validation sets (**A**,**B**). (**C**) shows efficacy of the Hazard Ratios (HRs) and 95% C.I. from proportional hazards models (Cox regression) to predict improved survival.

**Table 1 cancers-14-04012-t001:** Evaluation measures.

Problem Type	Measure	Explanation
Classification	AUC	Area under the receiver operating characteristic (ROC) curve. The ROC curve is created by plotting the true positive rate (TPR) against the false positive rate (FPR) at various discrimination threshold settings for binary classifiers.
	SE	Sensitivity or true positive rate (TPR), is the fraction of actual positive cases that has been predicted as positive in the population. It is an evaluation measure for binary classifiers.
	SP	Specificity or true negative rate (TNR) refers to the fraction of true negative that has been predicted as negative in the population. It is an evaluation measure for binary classifier.
	ACC	Accuracy is the ratio of the number of correct predictions to the total number of input samples.
	PPV	Positive predictive value is the ratio of patients truly diagnosed as positive to all those who had positive test results. It is an evaluation measure for binary classifiers.
	NPV	Negative predictive value refers to the fraction of subjects truly diagnosed as negative among those who had negative test results. It is an evaluation measure for binary classifiers.
	Classification Precision	Same as PPV
	F1 score	It is defined as the harmonic mean of the precision and recall
	PLR	The Positive Likelihood Ratio is defined as the ratio between the true positivity rate and the false positivity rate.
	NLR	The Negative Likelihood Ratio is the opposite of the PLR.
Regression\Survival Analysis	DFS	Disease-Free Survival is defined as the time from randomization to recurrence of tumor or death.
	TTNT	Time-to-next treatment is defined as the time between baseline (randomization, inclusion or treatment initiation) and the date of subsequent systemic treatment initiation.
	C-index	Harrell’s concordance C-index: used to evaluate risk models in survival analysis. It is computed as the fraction of concordant patient pairs on the sum of concordant and discordant patient pairs, where two patients are considered concordant if the patient with the higher risk score is the one having a shorter time-to-disease.
	HR	In survival analysis, the hazard ratio is defined as the ratio of the hazard rates corresponding to the conditions described by two levels of an explanatory variable (e.g., cases vs. controls)

**Table 2 cancers-14-04012-t002:** Artificial intelligence analysis of the response to standard chemotherapy alone for patients with colorectal metastasis. Data are grouped for imaging or molecular or clinical biomarkers based learning models.

Therapy	AI	Total, Training, Test	Input Features	Signature	Evaluated Models and Selected Model	Training Set *	Test Set *	Reference
** *Imaging biomarkers learning models* **								
First-line FOLFOX chemotherapy	ML	57 (242,172,70 features)	Contrast-enhanced computed tomography (CT) scans at differential time points 14 lesion shape-based 18 images intensity-based statistics 75 images gray level-based statistics	Delta-radiomics	Linear SVM Logistic regression Generalized linear Model with Poisson distribution Random Forest **Decision Tree**	Response vs. non-response: AUC = 0.99, 95% C.I. 0.97–1.0, 99% SE, 94% SP, 97% ACC, 95% PPV, 99% NPV	Response vs. non-response: AUC = 0.93, 95% C.I. 0.87–0.96, 85% SE, 92% SP, 86% ACC, 90% PPV, 97% NPV	[75]
First-line oxaliplatin-based regimens	ML	42 (126,94,32 features)	Contrast-enhanced CT scans at baseline	Radiomics		Response vs. non-response: AUC = 0.851, 95% C.I. 0.771–0.930	Response vs. non-response: AUC = 0.779, 95% C.I. 0.617–0.940	[79]
First-line CAPEOX or FOLFOX and FOLFIRI or XELIRI chemotherapies	DL	192,144,48 (3490 features)	Contrast-enhanced multidetector tomography (MDCT) scans at baseline; 583 radiomics features (traditional radiomics)	Fusion radiomics	**RESNET**	Response vs. non-response: AUC = 0.903, 95% C.I. 0.851–0.955, 84.7% SE, 84.8% SP, 84.7% ACC (radiomics model) Response vs. non-response: AUC = 0.935, 95% C.I. 0.897–0.973, 89.8% SE, 84.8% SP, 88.2% ACC (fusion radiomics model)	Response vs. non-response: AUC = 0.745, 95% C.I. 0.659–0.831, 90% SE, 73% SP, 85.4% ACC (radiomics model) Response vs. non-response: AUC = 0.830, 95% C.I. 0.688–0.973, 90.9% SE, 73.3% SP, 85.4% ACC (fusion radiomics model)	[80]
First-line FOLFOX or FOLFIRI chemotherapies	ML	92,54,38	CT scans at baseline 75 radiomics features	Radiomics	Gaussian Naive Bayes Multilayer Perceptron **Gaussian SVM**	Response vs. non-response: 76% SE, 67% SP, 72% ACC, 69%, PPV, 75% NPV (*per*-*lesion* 3D set)	Response vs. non-response: 61% SE, 60% SP, 61% ACC, 57% PPV, 64% NPV (*per*-*lesion* 3D set) Response vs. non-response: 41% SE, 21% SP, 32% ACC, 38% PPV, 23% NPV (*per*-*patient* 3D set)	[81]
First-line FOLFIRI chemotherapy	ML	129,78,51 (HQ set) 236,158,78 (SQ set)	High and standard quality (HQ and SQ sets) CT scans at differential time points 1749 Radiomics Features 1742 VGG16 extracted image features 8 rim specific features	Radiomics	For feature selection: RELIEF Fisher Score Chi Squared MRMR *t*-Test Wilcoxon Univariate models For classification: SVM KNN Naive Bayes Bagging Lasso **Random Forest**	Response vs. non-response: AUC = 0.75, 95% C.I. 0.63–0.85 (HQCT cohort) Response vs. non-response: AUC = 0.75, 95% C.I. 0.67–0.82 (SQCT cohort)	Response vs. non-response: AUC = 0.59, 95% C.I. 0.44–0.72 (HQCT cohort) Response vs. non-response: AUC = 0.55, 95% C.I. 0.43–0.66 (SQCT cohort)	[87]
** *Molecular biomarkers learning models* **								
First-line FOLFOX chemotherapy	ML	115,83,32	Microarray gene-expression data	74 genes	Iterative supervised learning (IML)	Response vs. non-response: 97.6% SE, 100% SP	Response vs. non-response: OS = 13.4 vs. 36.6 mo HR_OS_ = 2.6	[95]
First-line 5-fluorouracil-based regimens	ML	2317,2277,40	Immune infiltration data	16 long noncoding RNA	Feature selection: Ensemble of statistical and machine learning models Survival analysis: Cox proportional hazard model		Response vs. non-response: AUC = 0.843	[94]
First-line 5-fluorouracil-based regimens	ML	2307,2277,30	Immune infiltration data	16 long noncoding RNA	Feature selection: Ensemble of statistical and machine learning models Survival analysis: Cox proportional hazard model		Response vs. non-response: AUC = 0.765	[94]
First-line 5-fluorouracil-based regimens	ML	2201,2277,124	Immune infiltration data	16 long noncoding RNA	Feature selection: Ensemble of statistical and machine learning models Survival analysis: Cox proportional hazard model		Response vs. non-response: AUC = 0.709	[94]
First-line FOLFIRI chemotherapy	ML	82,27,55	Next generation sequencing (NGS) data	67 genes	SVM Logistic regression KNN Multilayer Perceptron Naive Bayes Quadratic Discriminant analysis Gaussian Process **Random Forest Ensemble**	Decreased survival: HR_OS_ = 2.631, 95% C.I. 1.041–6.649, OS = 18.7 vs. 34.4 mo		[99]
First-line FOLFOX chemotherapy	ML	83,54,29	Microarray differentially expressed (DE) gene profiles	18 genes	SVM KNN Gradient Boosting Machine **Random Forest** Decision Tree Neural Networks		Response vs. non-response: AUC = 0.877, 95% C.I. 0.747–1.00, 85% SE, 69.2% SP Improved survival: HR_OS_ = 0.358, 95% C.I. 0.178–0.717 in up-regulation of *MLK1* Improved survival: HR_OS_ = 0.563, 95% C.I. 0.336–0.943 in up-regulation of *CCDC124*	[101]
First-line FOLFIRI chemotherapy	ML	75,54,21	Microarray DE gene profiles	18 genes	SVM **Neural Networks** Random Forest		Response vs. non-response: AUC = 0.778, 95% C.I. 0.575–979	[101]
First-line oxaliplatin or irinotecan-based regimens	ML	176,99,77	Cytokine expression data	17 cytokines	Feature selection: SVM Classification: Multiple linear regression	Response vs. non-response: 83.5% SE, 80% SP, 81% ACC, OS = 18.4 vs. 51.5 mo	Response vs. non-response: 83.1% SE, 66.7% SP, 74.9% ACC), OS = 16.8 vs. 55.9 mo	[83]
First-line FOLFOX chemotherapy	ML	83,54,29	Microarray DE gene profiles	14 genes	Random Forest for feature selection, outlier detection and classification	Response vs. non-response: 91.3% SE, 95.6% SP, 80.2% ACC	Response vs. non-response: 80% SE, 92.8% SP, 69.2% ACC	[82]
First-line FOLFOX chemotherapy	ML	44	Protein pattern data	6-proteins		Response vs. non-response: 92.9% SE, 81.3% SP		[84]
First-line FOLFIRI chemotherapy	ML	26	Protein pattern data	7-proteins		Response vs. non-response: 92.3% SE, 92.3% SP		[84]
First-line FOLFIRI chemotherapy	ML	21	Microarray DE gene profiles	14-genes		Response vs. non-response: 92% SE, 100% SP, 95% ACC		[92]
	** *Clinical biomarkers learning models* **							
Irinotecan plus FOLFIRI chemotherapy	ML	20	Demographic data Liver function tests Tumor markers Pharmacokinetic parameters Anticancer treatment information; i.e., prior nonfuoropyrimidine regimen, prior surgery, prior radiation, fuoropyrimidine-based chemotherapy, concurrent surgery, concurrent radiation (categorical variables)	3pharmacoparameters	Classification: **BSLR: Backward Stepwise Logistic** **Regression (diarrhea)** C4.5 algorithm **Random Forest (leukopenia)** **SVM (neutropenia)**	Toxic side-effect: AUC = 0.74, 89% SE, 60% SP, 76% ACC (leukopenia) Toxic side-effect: AUC = 0.88, 70% SE, 70% SP, 75% ACC (neutropenia) for toxic side-effect: AUC = 0.95, 81% SE, 100% SP = 100%, 91% ACC (diarrhea)		[93]
First-line fluoropyrimidine-based regimens	ML	36,030 patients (training set size not reported, used data from 2006-11 + 10-fold cross validation) (test set size not reported, used data from 2012-14)	Anticancer treatment information; i.e., prior nonfuoropyrimidine regimen, prior surgery, prior radiation, fuoropyrimidine-based chemotherapy, concurrent surgery, concurrent radiation -cancer information; i.e., cancer stage, tumor sizes demographic variables; i.e., age, sex, race, and geographic areas socioeconomic status; i.e., county-level median household income, education prior cardiovascular disease histories and other comorbidities; i.e., hypertension, hyperlipidemia, diabetes prior medications; i.e., beta-blockers, angiotensin-converting enzyme (ACE) inhibitors		Logistic Regression Random Forest **XGBoost**		Toxic side-effect: precision = 0.621, F1 = 0.396 AUC = 0.801, 95% C.I. 0.781–0.821 (cardiotoxicity)	[96]

AI, artificial intelligence; ML, machine learning; DL, deep learning. AUC, receiver operating characteristic curve; SE, sensitivity; SP, specificity; ACC, accuracy; PPV, positive predictive value; NPV, negative predictive value; HR, hazard ratio.

**Table 3 cancers-14-04012-t003:** Artificial Intelligence analysis of the response to targeted therapy for patients with colorectal metastasis. Data are grouped for imaging or biomarkers based learning models.

Therapy	AI	Total, Training, Test	Input Features	Signature	Evaluated Models and Selected Model	Training Set	Test Set	Reference
** *Imaging biomarkers learning models* **								
Second-line anti-VEGF therapy (aflibercept plus FOLFIRI, prior bevacizumab plus oxaliplatin-based regimens)	DL	1028,502,526 (3757,1864,1893 lesions)	Automatically extracted features from CT scans at differential time points	Radiomics	GoogleNet + LSTM with 4 times steps	Response vs. non-response: C-Index = 0.678, 95% 0.650–0.706 (DL model)	Response vs. non-response: C-Index_OS_ = 0.649, 95% C.I. 0.619–0.679 (DL model) C-Index_OS_ = 0.694, 95% C.I. 0.661–0.720 (DL model) OS = 18 vs. 10.4 mo (HR = 0.49, 95% C.I. 0.40–0.61)	[85]
First-line anti-VEGF therapy (bevacizumab plus oxaliplatin/fluoropyrimidine-based regimens)	ML	76,52,24	Magnetic resonance imaging (MRI) at baseline 48 texture features 15 morphological features	Radiomics	Features Selection: Lasso Classification:LDA SVM KNNArtificial neural network Decision tree	*RAS* mutation status and non-responder to anti-EGFR therapy: AUC = 0.79, 95% C.I. 0.70–0.85, 78% SE, 74.2% SP, 76.1% ACC (texture and morphological features model) Response vs. non-response according to *RAS* mutation status: AUC = 0.84, 95% C.I. 0.780–0.91, 90% SE, 67.8% SP, 76.9% ACC (texture features model)	*RAS* mutation status and non-responder to anti-EGFR therapy: 83.3% SE, 75% SP, 79.2% ACC (texture and morphological features model) 91.7% SE, 83.3% SP, 87.5% ACC (texture features model)	[91]
First-line anti-VEGF therapy (bevacizumab with or without OLFOX/FOLFIRI/XELOX, combined with anti-EGFR in *KRAS*, *NRAS* and *BRAF* wild-types)	DL	180,101,79 (433,264,169 lesions)	MRI at differential time points	Radiomics	DC3CNN		Response vs. non-response: AUC = 0.849, 95% C.I. 0.737–0.926, 91.7% SE, 75% SP, ACC of 87.5%, 75% NPV, 91.7% PPV (four features model) Response vs. non-response: AUC = 0.833, 95% C.I. 0.695–1.000, 91.9% SE, 75% SP, ACC of 88.5%, 69.2% NPV, 93.8% PPV (four features model)	[98]
\First-line anti-VEGF therapy (bevacizumab plus FOLFOX chemotherapy)	DL	202,162,40 (treated/untreated lesion classification) 120,84,12,24 (treatment response classification)	Contrast-enhanced (CT) scans at baseline Texture analysis (TA) features	Radiomics	Treated/untreated classification: DT, SVM-RBF, ANN, **Inception-inspired CNN** Treatment response prediction: **CNN**	Response vs. non-response: AUC = 0.83, 95% C.I. 0.78–0.87, 97% SE, 59% SP, 78% ACC)	Response vs. non-response: AUC = 0.88, 95% C.I. 0.85–0.94, 98% SE, 54% SP, 76% ACC	[86]
First-line anti-EGFR therapy (cetuximab plus FOLFIRI chemotherapy)	ML	116,78,38 (HQ set) 186,124,62 (SQ set)	High and standard quality (HQ and SQ) CT scans at differential time points 1749 Radiomics Features 1742 VGG16 extracted image features 8 rim specific features	Radiomics	For feature selection: RELIEF Fisher ScoreChi Squared MRMR *t*-Test Wilcoxon Univariate models For classification: SVM KNN Naive Bayes Bagging Lasso **Random Forest**	Response vs. non-response: AUC = 0.83, 95% C.I. 0.75–0.92, 77% SE, 85% SP (HQ cohort) AUC = 0.84, 95% C.I. 0.76–0.89 (SQ cohort)	Response vs. non-response: AUC = 0.80, 95% C.I. 0.69–0.94, 80% SE, 78% SP (HQ cohort) AUC = 0.72, 95% C.I. 0.59–0.83, 82% SE, 61% SP (SQ cohort)	[87]
First-line anti-EGFR therapy (cetuximab plus FOLFIRI or FOLFOX chemotherapies)	ML	1886	CT scans and MRI at baseline	Radiomics	Clustering: K-Means	Response vs. non-response in *KRAS* mutated tumors: HR_OS_ = 1.44, 95% C.I. 1.08–1.92		[88]
Last-line dual anti-HER2 therapy (lapatinib plus trastuzumab or pertuzumab plus trastuzumabemantansine)	ML	38,28,10 patients 141,108,33 lesions	CT scans at baseline 24 radiomics features	Radiomics	Features Selection Genetic algorithms (GAs) ClassificationGaussian naïve Bayesian classifier	Response vs. non-response in *RAS* wild-type and HER2 amplified tumors: 89% SE, 85% SP, 93% PPV, 78% NPV (lesion model)	Response vs. non-response in *RAS* wild-type and HER2 amplified tumors: 90% SE, 42% SP, 73% PPV, 71% NPV (lesion model) 92% SE, 86% SP, 96% PPV, 75% NPV (patient model)	[74]
** *Molecular biomarkers learning models* **								
First-line anti-VEGF therapy (bevacizumab plus 5-fluorouracil-based regimens)	ML	2289,2277,12	Immune infiltration data	16-long noncoding RNA	Feature selection: Ensemble of statistical and machine learning models Survival analysis: Cox proportional hazard model		Response vs. non-response: AUC = 0.771	[94]
First-line anti-VEGF therapy (bevacizumab plus 5-fluorouracil-based regimens)	ML	2291,2277,14	Immune infiltration data	16-long noncoding RNA	Feature selection: Ensemble of statistical and machine learning models Survival analysis: Cox proportional hazard model		Response vs. non-response: AUC = 0.696	[94]
First-line anti-VEGF therapy (bevacizumab plus 5-fluorouracil-based regimens)	ML	2305,2277,28	Immune infiltration data	16-long noncoding RNA	Feature selection: Ensemble of statistical and machine learning models Survival analysis: Cox proportional hazard model		Response vs. non-response: AUC = 0.781	[94]
First-line anti-VEGF therapy (bevacizumab plus FOLFOX chemotherapy)	ML	650,105,545	Next generation sequencing (NGS) data	67-gene	SVM Logistic regression KNN Multilayer Perceptron Naive Bayes Quadratic Discriminant analysis Gaussian Process **Random Forest Ensemble**	Improved survival: HR_TTNT_ = 0.537, 95% C.I. 0.428–0.674 TTNT = 11.5 vs. 8.2 mo Improved survival: HR_OS_ = 0.466, 95% C.I. 0.325–0.670 OS = 42 vs. 24.5 mo	Improved survival: HR_OS_ = 0.629, 95% C.I. 0.404–0.981 OS = 30 vs. 15.9 mo	[99]
First-line anti-VEGF therapy (bevacizumab plus FOLFOXIRI chemotherapy)	ML	208,105,103	NGS data	67-gene	SVMLogistic regression KNN Multilayer Perceptron Naive Bayes Quadratic Discriminant analysis Gaussian Process **Random Forest Ensemble**		Improved survival: HR_OS_ = 0.483, 95% C.I. 0.270–0.864 OS = 30 vs. 15.9 mo	[99]
First-line anti-VEGF therapy (bevacizumab plus FOLFIRI chemotherapy	ML	488,345,143	Genotyping data	27 SNPs	Feature selection: **Random Survival Forests** Variable importance Minimal depth Survival analysis: Kaplan–Meier curves Log-rank test	Survival in *KRAS* wild-type with *CBP* rs129963 T/T variant: OS = 22.8 vs. 26 mo and PFS = 9.5 vs. 10.5 mo Survival in *KRAS* wild-type with TBK1 rs7486100 A/A variant: OS = 31.3 vs. 24.8 mo and PFS = 11.3 vs. 10.3 mo Survival in *CCL2* rs4586 T/T carriers: OS = 30.9 vs. 22.8 mo Survival in *VEGFR2* rs2305948 any C carriers: OS = 26.2 vs. 17.0 mo Survival in *DMRT1* rs755383 any T carriers: PFS = 9.4 vs. 9.0 mo Survival in *MMP2* rs243865 any T carriers: OS = 28.5 vs. 20.3 mo	Survival in *KRAS* mutant with β-catenin rs3864004 A/A genotype: OS = 16.3 vs. 26.3 mo and PFS = 7.8 vs. 9.6 mo Survival in *KRAS* mutant with *TBK1* rs7486100 A/A variant: PFS = 10.3 vs. 8.6 mo	[102]
First-line anti-VEGF therapy (bevacizumab plus FOLFIRI chemotherapy)	ML	558,180,378	Exome-sequencing data	1 or 2-SNPs	Combination of two Cox penalized regression models: Lasso Elastic Net	Improved survival: HR_PFS_ = 0.52, 95% C.I. 0.33–0.83 Decreased survival: HR_PFS_ = 2.3, 95% C.I. 1.19–4.57 Decreased survival in *NLRP1* any A and *SRL* AA carriers: HR_PFS_ = 8.3, 95% C.I. 3.3–21 and 2.2, 95% C.I. 1–5 Decreased survival in *KRAS* wild-type and concomitant carriers in combination with *NLRP1* any A and *SRL* AA: HR_PFS_ = 8.3, 95% C.I. 3.3–21 and 2.2, 95% C.I. 1–5 OS = 12 vs. 27 mo	Improved survival: HR_PFS_ = 0.42, 95% C.I. 0.21–0.85 in the *NLRP1* TT carriers Decreased survival: HR_PFS_ = 2.5, 95% C.I. 1.12–5.5 in the *SRL* AA carrier	[89]
Second-line anti-EGFR therapy (panitumumab with or without irinotecan, prior fluoropyrimidine-based regimens, with or without oxaliplatin)	ML	499,274,225	Immunochemistry (IHC) data	*Amphiregulin* /*epiregulin*		Improved survival in *KRAS* wild-type: HR_PFS_ = 0.54, 95% C.I. 0.37–0.79, OS = 8.0 vs. 3.2 mo Improved survival in *KRAS* and *BRAF* wild-types: HR_PFS_ = 0.53, 95% C.I. 0.36–0.78		[97]
First-line anti-EGFR therapy (CAPEOX-B chemotherapy plus or without cetuximab)	ML	553,368,185	Genome wide genotyping and survival data	781-SNPs	Modified Random Forest using SurvDiff in place of Gini index to split the data at each node Classic RF with survival derived data labels	Response vs. non-response: HR_benefit_ = 0.69, 95% C.I. 0.49–0.98 Response vs. non-response: HR_no benefit_ = 1.32, 95% C.I. 1.07–1.62 (SNP model) Response vs. non-response: HR_benefit_ = 0.52, 95% C.I. 0.35–0.76 (sex chromosome model)		[90]
First-line anti-EGFR therapy (FOLFIRI) or anti-VEGF therapy (fluoropyrimidine-based regimens, with or without oxaliplatin or irinotecan or FOLFOXIRI or other regimens plus or without bevacizumab)	ML	859,471,388	NGS data	7-gene classifier			Decreased survival: HR_PFS_ = 16.9, 95% C.I. 4.2–68.0	[100]

AI, artificial intelligence; ML, machine learning; DL, deep learning. AUC, receiver operating characteristic curve; SE, sensitivity; SP, specificity; ACC, accuracy; PPV, positive predictive value; NPV, negative predictive value; HR, hazard ratio; TTNT, time-to-next treatment.

**Table 4 cancers-14-04012-t004:** Least frequently met study quality items modified by Brnabic and Hess [72].

Study						
	State if Outliers with Impossible or Extreme Responses Are Removed; State Any Criteria Used for Outlier Removal.	State How Missing Values Were Handled.	External Validation Should also Be Performed Whenever Possible	If Possible, Report the Parameter Estimates in the Model and Their Confidence Intervals or Report Non-Parametric Estimates from Bootstrap Samples.	Meta-Analysis Inclusion	References
Johnson (2022)	Yes	Yes	Yes	No	No	[100]
Li (2022)	Yes	Yes	Yes	Yes	No	[96]
Liu (2022)	No	No	Yes	No	No	[94]
Giannini (2022)	Yes	Yes	Yes	Yes	Yes	[75]
Granata (2021)	Yes	Yes	Yes	Yes	No	[91]
Abraham (2021)	No	No	Yes	Yes	Yes	[99]
Naseem (2021)	No	No	Yes	No	No	[102]
Defeudis (2021)	Yes	No	No	No	No	[81]
Nakanishi (2021)	Yes	Yes	No	Yes	Yes	[79]
Wei (2021)	No	No	No	No	Yes	[80]
Williams (2021)	No	No	No	Yes	No	[97]
Tian (2021)	Yes	Yes	Yes	Yes	No	[95]
Lu (2021)	No	No	No	Yes	No	[85]
Giannini (2020)	Yes	Yes	No	No	No	[74]
Ubels (2020)	No	No	No	Yes	Yes	[90]
Dercle (2020)	No	Yes	No	Yes	Yes	[87]
Barat (2020)	No	No	Yes	Yes	No	[89]
Zhu (2020)	No	Yes	Yes	Yes	Yes	[98]
Maaref (2020)	No	No	No	Yes	Yes	[86]
Lu (2020)	No	No	No	Yes	Yes	[101]
Vera-Yunca (2020)	Yes	No	No	Yes	No	[88]
Oyaga-Iriarte (2019)	No	No	No	No	No	[93]
Chen (2014)	No	No	No	Yes	No	[83]
Tsuji (2012)	Yes	Yes	No	No	No	[82]
Yuan (2012)	No	No	No	No	No	[84]
Del Rio (2007)	No	No	No	Yes	No	[92]

**Table 5 cancers-14-04012-t005:** The 21 artificial intelligence models reporting the estimates of receiver operating characteristic curve (AUCs) and/or hazard ratios (HRs) and 95% Confidence Intervals (C.I.) for evaluating predictive response or overall survival included in the meta-analysis. Classifiers, cytotoxic chemotherapy, targeted therapy, population size, AUCs, HRs and 95% C.I. in training and/or validations sets are reported.

Study Included, Signature and Regimens		Performance Estimates	References
*Training sets*	*n*	*AUC, 95% C.I.*	
Giannini (2022), radiomics signature and chemotherapy	172	0.99, 95% C.I. 0.97–1.00	[75]
Nakanishi (2021), radiomics signature and chemotherapy	94	0.851, 95% C.I. 0.771–0.93	[79]
Wei (2021), radiomics signature and chemotherapy	144	0.935, 95% C.I. 0.897–0.973	[80]
Dercle (2020), radiomics signature and chemotherapy	78	0.75, 95% C.I. 0.63–0.85	[87]
Dercle (2020), radiomics signature and targeted therapy	78	0.83, 95% C.I. 0.75–0.92	[87]
Maaref (2020), radiomics signature and targeted therapy	162	0.83, 95% C.I. 0.78–0.87	[86]
*Validation sets*		*AUC, 95% C.I.*	
Giannini (2022), radiomics signature and chemotherapy	70	0.93, 95% C.I. 0.87–0.96	[75]
Nakanishi (2022), radiomics signature and chemotherapy	32	0.779, 95% C.I. 0.617–0.94	[79]
Wei (2021), radiomics signature and chemotherapy	48	0.830, 95% C.I. 0.688–0.973	[80]
Lu (2020), 18-gene signature and chemotherapy (FOLFOX)	29	0.877, 95% C.I. 0.747–1.00	[101]
Lu (2020) 18-gene signature and chemotherapy (FOLFIRI)	21	0.778, 95% C.I. 0.575–0.979	[101]
Dercle (2020), radiomics signature and chemotherapy	51	0.59, 95% C.I. 0.44–0.72	[87]
Dercle (2020), radiomics signature and targeted therapy	38	0.80, 95% C.I. 0.69–0.94	[87]
Zhu (2020), radiomics signature and targeted therapy	79	0.849, 95% C.I. 0.737–0.926	[98]
Zhu (2020), radiomics signature and targeted therapy	73	0.833, 95% C.I. 0.695–1.00	[98]
Maaref (2020), radiomics signature and targeted therapy	40	0.88, 95% C.I. 0.85–0.94	[86]
*Validation sets*		*HR, 95% C.I.*	
Lu (2020), *MLK1*-gene signature and chemotherapy (FOLFOX)	29	0.358, 95% C.I. 0.178–0.717	[101]
Lu (2020), *CCDC124*-gene signature and chemotherapy (FOLFOX)	29	0.563, 95% C.I. 336–0.943	[101]
Lu (2021), radiomics learning models and targeted therapy	526	0.49, 95% C.I. 0.4–0.61	[85]
Abraham (2021), 67-gene signature and targeted therapy	103	0.483, 95% C.I. 0.270–0.864	[99]
Abraham (2021), 67-gene signature and targeted therapy	545	0.629, 95% C.I. 0.404–0.981	[99]

**Table 6 cancers-14-04012-t006:** Performances of the artificial intelligence studies.

Therapy	AI	Signature	Training Set	Test Set	Reference
Chemotherapy	ML	Delta-radiomics	Excellent, 99% correctly classified as responders, 94% correctly classified as non-responders	Excellent, 85% correctly classified as responders, 92% correctly classified as non-responders	[75]
	ML	Radiomics	Good	Poor	[79]
	DL	Fusion radiomics	Excellent, 84.7% correctly classified as responders, 84.8% correctly classified as non-responders (radiomics model) Excellent, 89.8% correctly classified as responders, 84.8% correctly classified as non-responders (fusion radiomics model)	Poor, 90% correctly classified as responders, 73% correctly classified as non-responders (radiomics model) Good, 90.9% correctly classified as responders, 73.3% correctly classified as non-responders (fusion radiomics model)	[80]
	ML	Radiomics	76% correctly classified as responders, 67% correctly classified as non-responders (*per*-*lesion* 3D set)	61% correctly classified as responders, 60% correctly classified as non-responders (*per*-*lesion* 3D set) 41% correctly classified as responders, 21% correctly classified as non-responders (*per*-*patient* 3D set)	[81]
	ML	Radiomics	Poor (high standard quality computed tomography scan, HQCT, set) Poor (high standard quality computed tomography scan, SQCT, set)	Failed (HQCT set) Failed (SQCT set)	[87]
	ML	74 genes	97.6% correctly classified as responders, 100% correctly classified as non-responders		[95]
	ML	16 long noncoding RNA		Good	[94]
	ML	16 long noncoding RNA		Poor	[94]
	ML	16 long noncoding RNA		Poor	[94]
Targeted therapy	DL	Radiomics		Good, 91.7% correctly classified as responders, 75% correctly classified as non-responders (four features model) Good, 91.9% correctly classified as responders, 75% correctly classified as non-responders (four features model)	[98]
	DL	Radiomics	Good, 97% correctly classified as responders, 59% correctly classified as non-responders	Good, 98% correctly classified as responders, 54% correctly classified as non-responders	[86]
	ML	Radiomics	Good, 77% correctly classified as responders, 85% correctly classified as non-responders (high quality, HQ, cohort) Good (standard quality, SQ, cohort)	Good, 80% correctly classified as responders, 78% correctly classified as non-responders (HQ cohort) Poor, 82% correctly classified as responders, 61% correctly classified as non-responders (SQ cohort)	[87]
	ML	Radiomics	89% correctly classified as responders, 85% correctly classified as non-responders (lesion model)	90% correctly classified as responders, 42% correctly classified as non-responders (lesion model) 92% correctly classified as responders, 86% correctly classified as non-responders (patient model)	[74]
	ML	16-long noncoding RNA		Poor	[94]
	ML	16-long noncoding RNA		Worthless	[94]
	ML	16-long noncoding RNA		Poor	[94]
	ML	67-gene	Good	Poor	[99]
Chemotherapy and targeted therapy	ML/DL	Imaging/molecular signatures	Good	Good	Present meta-analysis

ML, machine learning; and DL, deep learning.

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
