# Peer review of "Artificial Intelligence Predictive Models of Response to Cytotoxic Chemotherapy Alone or Combined to Targeted Therapy for Metastatic Colorectal Cancer Patients: A Systematic Review and Meta-Analysis"

_cancers, 2022, doi:10.3390/cancers14164012_

Round 1

Reviewer 1 Report

Summary:

 Thank you for the opportunity to review this paper.

This is a systematic review and meta-analysis of existing literature applying AI techniques to predict treatment response in metastatic colorectal cancer.

The given aim is to: “assess the effectiveness of learning algorithms to predict the response to chemotherapy alone or combined to targeted therapy in this patient setting…with the final goal of evaluating whether learning models could be used in clinical trials to assess prognosis or toxicity in standard of care settings and as predictors of therapy response”.

Table 2 and 3 provide useful grouping of published studies and the ensuing section provides in-depth reporting of individual study results.

Table 4 provides a summary of the study data into in the meta-analysis and Figures 2A,B and C summarise the meta-analysis results.

Summarising this diverse and detailed literature is challenging and the authors are to be commended on their efforts.

Major questions and comments:

1.     The premise provided in the introduction is “Evolution of genetically and/or epigenetically diverse tumor-cell populations during tumor growth and progression is considered the main factor causing tumor heterogeneity, displaying inherent functional variability in tumor propagation potential and tolerance to pharmacological treatment with either cytotoxic or targeted cancer therapeutics.” Given that it is the tumour cell profile that is considered the main factor in treatment response or resistance, it is not a huge leap of logic to see why genetic and molecular biomarkers are included as inputs for predictive models. However, it is less clear, based on the given premise, why radiomics inputs are included in predictive models.

Please provide the rationale (preferably biological rationale) for inclusion of radiomics as inputs for treatment prediction.

2.     The authors state that: “Therefore, there is an unmet need of early identification of mCRC cases who will be responsive to specific regimens and clinical indicator of toxicity burden.”

Given this context and that the stated goal is eventual translation into clinical trials and then practice, it would be useful to contextualise the data available for early identification of treatment response. For example, what data is routinely collected in clinical practice that could be used for clinical decision making (e.g. CTs, MRIs, tumour molecular and genetic mutations). Are the data inputs used in these studies routinely available in clinical practice today?

3.     The authors rightly state that: “In ML/DP, the quality of the datasets is fundamental and a preprocessing strategy, encompassing feature standardization and extraction is a necessary step before applying any learning algorithm.”

How was study data quality assessed in the review and meta-analysis? What impact did data quality have on the results?

4.     If available, I think the reader would benefit from more information about the sampling of the testing and training data sets.

For example:

-       Many of the studies had extremely small sample sizes. Please include comment in the limitations section about the impact of these small samples sizes on the study outcomes and whether or not this impacts the validity of the overall results 

-        How many images per person were used in the training and testing data sets for radiomics studies?

-        Were data shared between training and testing data sets in any of the studies?

-        How were outcome cases distributed in the testing and training data sets (i.e. were there equal numbers of responders vs non-responders etc)

5.     Please include more information regarding the outcome of the studies. For example, how was response or non-response defined? Over what time period was response assessed and was this period consistent between studies?

6.     Table 2 & 3 comments and questions:

-        Table 2 & 3 include many useful features including the models inputs. Please also include the output being trained (e.g. response vs non-response) for each study

-        Do the performance metrics relate to response vs no response? For example, is the PPV predicting response or non-response?

-        For some studies, inputs are described as from differential timepoints. Please clarify what is meant by this (i.e. are the inputs from different timepoints in the clinical journey for different people or are they from different timepoints in the same person, or something else?)

7.     Discussion: I think it is important to provide a rational framework for interpreting model performance in the clinical setting. In the discussion certain models are described as having ‘high’ performance without the criteria for performance being defined (specifically for clinical practice). An AUC of 0.903 is generally good model performance in ML contexts, but I would argue that it is certainly not ‘high’ enough for clinical applications. 10% misclassification may have significant implications in the clinical space.

Please provide a rationale for classification of performance as ‘high’ or poor in the methods section of the paper. Please include a plain language explanation of the interpretation of these classifications to aid clinical readers to translate the findings. e.g. XX % correctly classified as non-responders or XX% correctly classified as responders.

8.     The discussion would benefit from further contextualisation regarding the gap that these studies are trying to fill. For example, why are ML techniques required? What are the failures in current clinical processes and guidelines for selecting chemotherapy and other regimes based on molecular tumour profiles? Why is it clinically important to distinguish between responders and non-responders? Are there alternative therapies that can be used instead?

9.     The discussion would benefit from further clinical contextualisation about how the performance of these models compared to current clinical guidelines. For example, what proportion of mCRC patients are currently offered a treatment that is unlikely to be effective for them and is the performance of ML/DL models comparable, better or worse than current clinical decision making?

10.   Finally, I think it would be helpful for the reader if the authors were to specify what the scope of translation might look like in the future, especially with regard to clinical trials. For example, are the authors suggesting that patients would be randomised to treatment based on a predictive model vs usual treatment, and if so, are the models appropriately advanced to make such a trial ethical?

Minor comments:

Numbers are sometimes separated with a full stop instead of a comma e.g. “The final database consisted of 25 studies with 7.762 patients eligible for inclusion in the database”. Should this be 7,762 patients?

Author Response

Point by point response  

Reviewer N.1

Major questions and comments:

The premise provided in the introduction is “Evolution of genetically and/or epigenetically diverse tumor-cell populations during tumor growth and progression is considered the main factor causing tumor heterogeneity, displaying inherent functional variability in tumor propagation potential and tolerance to pharmacological treatment with either cytotoxic or targeted cancer therapeutics.” Given that it is the tumour cell profile that is considered the main factor in treatment response or resistance, it is not a huge leap of logic to see why genetic and molecular biomarkers are included as inputs for predictive models. However, it is less clear, based on the given premise, why radiomics inputs are included in predictive models.

Question N1. Please provide the rationale (preferably biological rationale) for inclusion of radiomics as inputs for treatment prediction.

Answer: We thank the Reviewer for his/her question. The rationale for inclusion of radiomics was provided in the Introduction (last paragraph):

Introduction

“Then, accumulating evidence suggests that radiomics can be used to study tumor heterogeneity and to predict therapy in CRC [25]. Biological insight on data-driven radiomics features has been recently provided by different biological metrics such as gene expression, immunohistochemistry staining intensity and microscopic histologic textures [26]. In particular, radiomics in combination with molecular biomarkers have been even shown to be able of predicting genetic mutations including both KRAS mutations and KRAS/BRAF status in CRC patients [25].”

Question N.2 The authors state that: “Therefore, there is an unmet need of early identification of mCRC cases who will be responsive to specific regimens and clinical indicator of toxicity burden.” Given this context and that the stated goal is eventual translation into clinical trials and then practice, it would be useful to contextualise the data available for early identification of treatment response. For example, what data is routinely collected in clinical practice that could be used for clinical decision making (e.g. CTs, MRIs, tumour molecular and genetic mutations). Are the data inputs used in these studies routinely available in clinical practice today?

Answer: We thank the Reviewer for his/her question, we have added this information in the Introduction (at the 3° paragraph):

Introduction

 “Indeed, at the present, no imaging criteria or molecular biomarkers are currently available in clinical practice today for early identification of treatment response before the start of the therapy and even the radiologists cannot have sufficient imaging information on the baseline examination to predict which tumor lesions will respond to the treatments.”

Question N.3. The authors rightly state that: “In ML/DP, the quality of the datasets is fundamental and a preprocessing strategy, encompassing feature standardization and extraction is a necessary step before applying any learning algorithm.” How was study data quality assessed in the review and meta-analysis? What impact did data quality have on the results?

Answer: We thank the Reviewer for his/her question. We added two new Sections titled “2.4. Publication quality” and “3.2. Publication validation” to M&M and Results and two new Tables 4 and Table S1 to address this important aspect that it was also commented in the Discussion (at the 6° paragraph). We hope to have better clarified the results of our study.

2.4. Publication quality The study of publication quality was assessed by two investigators (MP and EL) using study quality items that were criterions of quality publication in the Luo scale [71], in keeping with a recent systematic review on ML studies [72].

3.2. Publication validation When data quality was assessed in the 26 AI study included in the systematic review and the meta-analysis [74, 75, 79-102], we identified a common gap that consisted in the lack of an external validation cohort, that was missing in more than the half of the investigations (Table 4, and Table 1 Supplementary). Also, outliers, missing values and C.I. were not handled by most of the studies.

Discussion

There are several limitations to this study. First, data of a same cohort were shared between training and test sets in remaining investigations, whereas the use of an external validation cohorts, that is critical to ensure model accuracy, was conducted only in the half of the publications. In addition, three studies that did not split the data [79,92,93]. This can be due to the lack of appropriate cohorts or the lack of awareness of the importance of an external validation cohort before its application in clinical settings. Missing and outliers were also not appropriately managed by most of the AI studies. Some of them have also small sample sizes that could be of impact on study outcomes and decrease the statistical power. Then, the size of the validation sets was generally lower than that observed in the training sets, possibly reflecting the low number of mCRC patients that were available overall. Nevertheless, the use of k-fold validation that allow proper stratification of predictive outcomes [94, 101] have been performed by the majority of the studies (77%), even at greater rate than previous review [72].

Question N.4 If available, I think the reader would benefit from more information about the sampling of the testing and training data sets. For example: Many of the studies had extremely small sample sizes. Please include comment in the limitations section about the impact of these small samples sizes on the study outcomes and whether or not this impacts the validity of the overall results 

Answer: see the answer to the question N.3.

Question N.5 How many images per person were used in the training and testing data sets for radiomics studies?

Answer: unfortunately, this information was not reported.

Question N.6 Were data shared between training and testing data sets in any of the studies?

Answer: In the AI investigations, cohorts were generally divided into a training set and a validation set to train and validate the developed models to avoid overfitting, with the exception of three studies [79,92,93], that did not split the data. Data on training and validation set were also not always reported. This was added in the Results, section 3.3. We hope to have better clarified the results of our study.

Question N.7 How were outcome cases distributed in the testing and training data sets (i.e. were there equal numbers of responders vs non-responders etc)

Answer: Information on the distribution of the outcome case, e.g. numbers of responders vs. non-responders, was generally missing, with the exception of three AI studies, where the following frequency were reported: 131 responder vs. 61 non-responders in the study of Wei et al. [80], 42 responder vs. 41 non responders in the study of Tian et al. [95], and 184 responders vs. 210 non responders in the study of Liu et al. [94]. This was added in the Results, section 3.3.

Question N.8 Please include more information regarding the outcome of the studies. For example, how was response or non-response defined? Over what time period was response assessed and was this period consistent between studies?

Answer: We thank the Reviewer for his/her question. A new section titled “3.1 Outcomes and performance estimates of predictive models” was added to the M&M section to report this information. We hope to have better clarified the results of our study.

3.1 Outcomes and performance estimates of predictive models

Different clinical endpoints, including the median values of the OS, the PFS, the TTNT or the Response Evaluation Criteria in Solid Tumors (RECIST) criteria, were used to analyze the response to therapy in periods of times consistent between different studies, commonly after 6 or more months from the last cycle of chemotherapy, but shorter time pharmacoparameters were employed to study toxic side-effects.

Question N.9 Table 2 & 3 comments and questions: Table 2 & 3 include many useful features including the models inputs. Please also include the output being trained (e.g. response vs non-response) for each study

Answer: output being trained was added in Tables 2 and 3

Question N.10 Do the performance metrics relate to response vs no response? For example, is the PPV predicting response or non-response?

Answer: We thank the Reviewer for his/her question. New sections titled “3.1 Outcomes and performance estimates of predictive models”, “3.9. Predictive model performances”, and the new Table 6 were added to the Results. Performance were commented in the Discussion, Conclusions and reported in the Abstract. We hope to have better clarified the results of our study.

3.1 Outcomes and performance estimates of predictive models

Several performance metrics, such as the AUC, the Harrell’s concordance C (C-index), the SE, the SP, the accuracy (ACC), the positive predictive value (PPV) and the negative predictive value (NPV), the classification precision and F1 scores (the harmonic means of the precision and recall) and patient survival estimation by hazard ratio (HR) were used to evaluate the predictive power of various algorithms for correctly classifying response vs. non-response.

3.9. Predictive model performances

Model performances were then analyzed in 13 AI studies that reported AUCs, SE and SP estimates [74,75,79-81,86,87,94,95,98,99,101]. Considering the AI studies that have analyzed the model performance using the AUCs, Table 6 shows that both radiomics and molecular signatures were effective in predicting the response to therapy with good or excellent estimates in both the training and the validations sets. Comparable estimates of the global AUCs were also reported by our meta-analysis. Moreover, when we evaluated the AI investigation with tests that reached the 80% SE and the 90% SP, there were the delta radiomics that showed the SE of 99% and the SP of 94% in the training set and the SE of 85% and the SP of 92 in the test set, and the 74 gene signatures that obtained the SE of 97.6% and the SP of 100% in the training set.

Table 6. Performances of the artificial intelligence studies.

Therapy

AI

Signature

Training set

Test set

Reference

Chemotherapy

ML

Delta-radiomics

Excellent, 99% correctly classified as responders, 94% correctly classified as non-responders

Excellent, 85% correctly classified as responders, 92% correctly classified as non-responders

[75]

ML

Radiomics

Good

Poor

[79]

DL

Fusion radiomics

Excellent, 84.7% correctly classified as responders, 84.8% correctly classified as non-responders (radiomics model)

Excellent, 89.8% correctly classified as responders, 84.8% correctly classified as non-responders (fusion radiomics model)

Poor, 90% correctly classified as responders, 73% correctly classified as non-responders (radiomics model)

Good, 90.9% correctly classified as responders, 73.3% correctly classified as non-responders (fusion radiomics model)

[80]

ML

Radiomics

76% correctly classified as responders, 67% correctly classified as non-responders (per-lesion 3D set)

61% correctly classified as responders, 60% correctly classified as non-responders (per-lesion 3D set)

41% correctly classified as responders, 21% correctly classified as non-responders (per-patient 3D set)

[81]

ML

Radiomics

Poor (high standard quality computed tomography scan, HQCT, set)

Poor (high standard quality computed tomography scan, SQCT, set)

Failed (HQCT set)

Failed (SQCT set)

[87]

ML

74 genes

97.6% correctly classified as responders, 100% correctly classified as non-responders

[95]

ML

16 long noncoding RNA

Good

[94]

ML

16 long noncoding RNA

Poor

[94]

ML

16 long noncoding RNA

Poor

[94]

Targeted therapy

DL

Radiomics

Good, 91.7% correctly classified as responders, 75% correctly classified as non-responders (four features model)

Good, 91.9% correctly classified as responders, 75% correctly classified as non-responders (four features model)

[98]

DL

Radiomics

Good, 97% correctly classified as responders, 59% correctly classified as non-responders

Good, 98% correctly classified as responders, 54% correctly classified as non-responders

[86]

ML

Radiomics

Good, 77% correctly classified as responders, 85% correctly classified as non-responders (high quality, HQ, cohort)

Good (standard quality, SQ, cohort)

Good, 80% correctly classified as responders, 78% correctly classified as non-responders (HQ cohort)

Poor, 82% correctly classified as responders, 61% correctly classified as non-responders (SQ cohort)

[87]

ML

Radiomics

89% correctly classified as responders, 85% correctly classified as non-responders (lesion model)

90% correctly classified as responders, 42% correctly classified as non-responders (lesion model)

92% correctly classified as responders, 86% correctly classified as non-responders (patient model)

[74]

ML

16-long noncoding RNA

Poor

[94]

ML

16-long noncoding RNA

Worthless

[94]

ML

16-long noncoding RNA

Poor

[94]

ML

67-gene

Good

Poor

[99]

Chemotherapy and targeted therapy

ML/DL

Imaging/molecular signatures

Good

Good

Present meta-analysis

Discussion

In this context, our meta-analysis demonstrated a good performance power of the learning models used to predict response to chemotherapy alone or combined to targeted therapy by discriminating response vs. non-response. Also, the calculation of overall HR shows that learning models have strong ability to predict improved survival. Next, when we have analyzed performances, both radiomics and molecular signatures showed to be effective to predict the response to therapy with good or excellent values. In particular, we found that both the delta radiomics and the 74 gene signatures were able to discriminate response vs. non-response by correctly classifying up to 99% of mCRC patients as responders and up to 100% of patients as non-responders.

(at 5° paragraph)

Conclusions

In addition, the delta-radiomics and the 74 gene signatures were found to able to discriminate response vs. non-response by correctly identifying up to 99% of mCRC patients as responders and up to 100% of patients as non-responders.

Abstract

Lastly, the delta-radiomics and the 74 gene signatures were able to discriminate response vs. non-response by correctly identifying up to 99% of mCRC patients as responders and up to 100% of patients as non-responders. Specifically, when we evaluated the predictive models with tests reaching the 80% sensitivity (SE) and the 90% specificity (SP), the delta radiomics showed the SE of 99% and the SP of 94% in the training set and the SE of 85% and the SP of 92 in the test set, whereas the 74 gene signatures the SE of 97.6% and the SP of 100% in the training set.

Question N.11 For some studies, inputs are described as from differential timepoints. Please clarify what is meant by this (i.e. are the inputs from different timepoints in the clinical journey for different people or are they from different timepoints in the same person, or something else?)

Answer: In radiomics studies, different timepoints in the same person were often used in both the training and testing sets, e.g. at baseline and after chemotherapy.

See also the answer to the Questions 6 and 7. 

Question N.12 Discussion: I think it is important to provide a rational framework for interpreting model performance in the clinical setting. In the discussion certain models are described as having ‘high’ performance without the criteria for performance being defined (specifically for clinical practice). An AUC of 0.903 is generally good model performance in ML contexts, but I would argue that it is certainly not ‘high’ enough for clinical applications. 10% misclassification may have significant implications in the clinical space.

Please provide a rationale for classification of performance as ‘high’ or poor in the methods section of the paper. Please include a plain language explanation of the interpretation of these classifications to aid clinical readers to translate the findings. e.g. XX % correctly classified as non-responders or XX% correctly classified as responders.

Answer: We thank the Reviewer for his/her question. A new section titled “3.5. Rationale framework” was added to Results. See also the answer to the Question N.9.

3.5. Rationale framework

A rational framework for interpreting model performances was developed using the area under the receiver operating characteristic (ROC) curve (AUC), and the sensitivity (SE) and the specificity (SP) estimates reported in the AI models. AUC values were interpreted as follows: 0.6-0.7 (worthless), 0.7-0.8 (poor), 0.8-0.9 (good), and > 0.9 (excellent) [73]. In addition, the SE was referred to the ability of a test to correctly identify patients who are responders, whereas the SP to the ability to correctly classify those who are non-responders [74, 75].

Question 13. The discussion would benefit from further contextualisation regarding the gap that these studies are trying to fill. For example, why are ML techniques required? What are the failures in current clinical processes and guidelines for selecting chemotherapy and other regimes based on molecular tumour profiles? Why is it clinically important to distinguish between responders and non-responders? Are there alternative therapies that can be used instead?

Answer: We thank the Reviewer for his/her question aimed at better clinically contextualize our work. Indeed, the failure in current clinical process is often not predictable despite the fact that the treatment choice appears to be optimal for that particular patient with that particular disease. It happens, in fact, that regardless of the correctness of the therapeutic choice a patient will experience disease progression earlier than expected ("non responder"). The analysis of a huge number of routinely available variables will make it possible to identify a certain characteristic of that specific patient/cancer that might be useful in avoiding future therapeutic failures. We have added a new paragraph at the beginning of the Discussion section and hope to have better clarified the context in which our study has arisen.

Discussion

Selection of the optimal first-line treatment represents a crucial step in the therapeutic pathway of mCRC patients, in order to obtain a significant improvement of PFS and, possibly, OS due to the development and combination of cytotoxic and biologic drugs chosen on the basis of the tumour mutational status. However, biological drugs, being directed towards specific "actionable" targets, might cause a heterogeneous tumour response, depending on clinical characteristics and/or disease biology of each patient. Therefore, despite selection of optimal therapy is based on the patient’s molecular phenotype, a percentage of patients is not responsive to targeted therapy, leaving to hypothesize that additional mediators could be involved in the dysregulation of molecular mechanisms, expression of further potentially actionable genes might intervene and/or that mechanisms of innate/acquired resistance to target inhibitors occur. Even the presence of genetic mutations in tumor RAS or BRAF sequences and MSI status cannot always predict the therapeutic response in mCRC patients [5]. No imaging criteria are also available that could predict the response to therapy before the start of the therapy and or even predict the kind metastatic lesions that will respond to the treatment [125]. To face these limits, the design of optimal strategy for mCRC on a case by case basis has been proposed, where therapeutic interventions should be modulated depending on patient/tumor’s characteristics. Therefore, it is relevant to characterize CRC complexity at individual level to understand if a patient will respond to a specific therapy or will show resistance, considering the opportunity to leverage new emerging computer science solutions from research to real-world conditions. In this regard, growing emphasis has been put on clinical decision support systems based on AI, in general, and ML techniques, in particular to develop predictive models of cancer progression or response to treatment using ML techniques, especially in mCRC. This approach has already proven capable of exploiting significant patterns in routinely collected demographic, clinical and biochemical data and allowed the design of clinical decision support systems (DSS) that can be easily adapted to different tumors [126, 127].

Question 14. The discussion would benefit from further clinical contextualisation about how the performance of these models compared to current clinical guidelines. For example, what proportion of mCRC patients are currently offered a treatment that is unlikely to be effective for them and is the performance of ML/DL models comparable, better or worse than current clinical decision making?

Answer: We thank the Reviewer for his/her question. A new paragraph (8°) was added to the Discussion.

Discussion

When resistance to treatments occurs, few treatment options are available for resistant mCRC that may be difficult to tackle through non-cross-resistant anticancer drugs for second and subsequent lines of therapy despite new diagnostic and therapeutic methods [10]. Currently, about 20-25% of mCRC patients with hepatic metastasis are resectable, 60–70% of distant mCRC patients will develop local or distant recurrence, while only 20% will achieve long-term remission [136]. Patients with a poor performance status represent a proportion of mCRC patients of up to 41% of routine care mCRC, who have more toxic side-effects if they are treated as well as inferior survival outcomes [137]. Furthermore, we cannot stratify mCRC patients or those who are at risk of developing colorectal metastasis at diagnosis using biomarkers or other currently available techniques. Even the presence of genetic mutations in tumor RAS or BRAF sequences and MSI status cannot always predict the therapeutic response in mCRC patients [5]. On the other hand, AI signatures can reach better performance in distinguish responders from non-responders in respect to routine clinical indicators, such as cancer stage, adjuvant therapies, surgery on primary tumor and RECIST criteria [5,80,83,85,98,100,138]. New classifications of mCRC based on AI parameters will soon emerge and we will be able to identify new subtypes of mCRC patients, for whom the definition will be based on combinations of radiomics and/or molecular signatures contributing to improve the clinical management of mCRC and clinical decision making. AI approach might provide innovative tools to distinguish between responders and non-responders, before treatment start, to know if a mCRC patient will respond to a specific therapy or will show resistance.

Question 15.  Finally, I think it would be helpful for the reader if the authors were to specify what the scope of translation might look like in the future, especially with regard to clinical trials. For example, are the authors suggesting that patients would be randomised to treatment based on a predictive model vs usual treatment, and if so, are the models appropriately advanced to make such a trial ethical?

Answer: We thank the Reviewer for his/her question. A new paragraph was added to the Discussion (the last one).

Discussion

In regard the scope of translation that might look like in the future, especially with regard to clinical trials, our findings suggest that mCRC patients would be randomized to treatment based on a predictive model vs. usual treatment. However, the possibility of employing predictive models in choosing the most effective and cost-effective therapeutic intervention for cancer patients relies on the development of improved and innovative AI based computational frameworks. These models must have been built on the basis of the evaluation of a huge amount of clinical data, with emphasis on factors influencing positive and negative responses, prognostic biomarkers, and molecular predictors of therapeutic response or disease to be appropriately advanced to make such trial ethically correct. A rigorous process must always underlie any AI model to ensure reliable prediction of mCRC patient response to treatments. In this field, we are awaiting the first results of the European Union funded study titled “Targeted therapy for advanced colorectal cancer patients, REVERT” (https://www.revert-project.eu/) aimed at strengthening this ambitious goal by addressing the specific challenge of understanding at system level the pathophysiology of mCRC cancer in patients responding well or poorly to therapies, to design optimal strategy for mCRC on a case by case basis, with therapeutic interventions modulated depending on patient’s features.

Minor comments:

Numbers are sometimes separated with a full stop instead of a comma e.g. “The final database consisted of 25 studies with 7.762 patients eligible for inclusion in the database”. Should this be 7,762 patients?

Answer: Numbers were corrected

Reviewer 2 Report

The authors present a comprehensive overview about the actual studies on the use AI for predicting the outcome of chemotherapy and immunotherapy for metastatic colorectal cancer. 

I think apart from really supporting the ideas and results of the different trials, the authors should also pint out, which studies could be seen critical:

- because the training results were not fulfille

- because the AUR is low

I guess this would be very interesting for the reader.

Author Response

Reviewer N.2

The authors present a comprehensive overview about the actual studies on the use AI for predicting the outcome of chemotherapy and immunotherapy for metastatic colorectal cancer. 

Question N.1 I think apart from really supporting the ideas and results of the different trials, the authors should also pint out, which studies could be seen critical: - because the training results were not fulfille

Answer: see the answer to the Question 6 and also to the Questions 3, 4 and 10 of the first reviewer.

Es. Question N.6 Were data shared between training and testing data sets in any of the studies?

Answer: In the AI investigations, cohorts were generally divided into a training set and a validation set to train and validate the developed models to avoid overfitting, with the exception of three studies [79,92,93], that did not split the data. Data on training and validation set were also not always reported. This was added in the Results, section 3.3. We hope to have better clarified the results of our study.

Es. Question N.3. The authors rightly state that: “In ML/DP, the quality of the datasets is fundamental and a preprocessing strategy, encompassing feature standardization and extraction is a necessary step before applying any learning algorithm.” How was study data quality assessed in the review and meta-analysis? What impact did data quality have on the results?

Answer: We thank the Reviewer for his/her question. We added two new Sections titled “2.4. Publication quality” and “3.2. Publication validation” to M&M and Results and two new Tables 4 and Table S1 to address this important aspect that it was also commented in the Discussion (at the 6° paragraph). We hope to have better clarified the results of our study.

2.4. Publication quality The study of publication quality was assessed by two investigators (MP and EL) using study quality items that were criterions of quality publication in the Luo scale [71], in keeping with a recent systematic review on ML studies [72].

3.2. Publication validation When data quality was assessed in the 26 AI study included in the systematic review and the meta-analysis [74, 75, 79-102], we identified a common gap that consisted in the lack of an external validation cohort, that was missing in more than the half of the investigations (Table 4, and Table 1 Supplementary). Also, outliers, missing values and C.I. were not handled by most of the studies.

Discussion

There are several limitations to this study. First, data of a same cohort were shared between training and test sets in remaining investigations, whereas the use of an external validation cohorts, that is critical to ensure model accuracy, was conducted only in the half of the publications. In addition, three studies that did not split the data [79,92,93]. This can be due to the lack of appropriate cohorts or the lack of awareness of the importance of an external validation cohort before its application in clinical settings. Missing and outliers were also not appropriately managed by most of the AI studies. Some of them have also small sample sizes that could be of impact on study outcomes and decrease the statistical power. Then, the size of the validation sets was generally lower than that observed in the training sets, possibly reflecting the low number of mCRC patients that were available overall. Nevertheless, the use of k-fold validation that allow proper stratification of predictive outcomes [94, 101] have been performed by the majority of the studies (77%), even at greater rate than previous review [72].

Es. Question N.4 If available, I think the reader would benefit from more information about the sampling of the testing and training data sets. For example: Many of the studies had extremely small sample sizes. Please include comment in the limitations section about the impact of these small samples sizes on the study outcomes and whether or not this impacts the validity of the overall results 

Answer: see the answer to the question N.3.

Es. Question N.10 Table 2 & 3 comments and questions: Do the performance metrics relate to response vs no response? For example, is the PPV predicting response or non-response?

Answer: We thank the Reviewer for his/her question. New sections titled “3.1 Outcomes and performance estimates of predictive models”, “3.9. Predictive model performances”, and the new Table 6 were added to the Results. Performance were commented in the Discussion, Conclusions and reported in the Abstract. We hope to have better clarified the results of our study.

3.1 Outcomes and performance estimates of predictive models

Several performance metrics, such as the AUC, the Harrell’s concordance C (C-index), the SE, the SP, the accuracy (ACC), the positive predictive value (PPV) and the negative predictive value (NPV), the classification precision and F1 scores (the harmonic means of the precision and recall) and patient survival estimation by hazard ratio (HR) were used to evaluate the predictive power of various algorithms for correctly classifying response vs. non-response.

3.9. Predictive model performances

Model performances were then analyzed in 13 AI studies that reported AUCs, SE and SP estimates [74,75,79-81,86,87,94,95,98,99,101]. Considering the AI studies that have analyzed the model performance using the AUCs, Table 6 shows that both radiomics and molecular signatures were effective in predicting the response to therapy with good or excellent estimates in both the training and the validations sets. Comparable estimates of the global AUCs were also reported by our meta-analysis. Moreover, when we evaluated the AI investigation with tests that reached the 80% SE and the 90% SP, there were the delta radiomics that showed the SE of 99% and the SP of 94% in the training set and the SE of 85% and the SP of 92 in the test set, and the 74 gene signatures that obtained the SE of 97.6% and the SP of 100% in the training set.

Table 6. Performances of the artificial intelligence studies.

Therapy

AI

Signature

Training set

Test set

Reference

Chemotherapy

ML

Delta-radiomics

Excellent, 99% correctly classified as responders, 94% correctly classified as non-responders

Excellent, 85% correctly classified as responders, 92% correctly classified as non-responders

[75]

ML

Radiomics

Good

Poor

[79]

DL

Fusion radiomics

Excellent, 84.7% correctly classified as responders, 84.8% correctly classified as non-responders (radiomics model)

Excellent, 89.8% correctly classified as responders, 84.8% correctly classified as non-responders (fusion radiomics model)

Poor, 90% correctly classified as responders, 73% correctly classified as non-responders (radiomics model)

Good, 90.9% correctly classified as responders, 73.3% correctly classified as non-responders (fusion radiomics model)

[80]

ML

Radiomics

76% correctly classified as responders, 67% correctly classified as non-responders (per-lesion 3D set)

61% correctly classified as responders, 60% correctly classified as non-responders (per-lesion 3D set)

41% correctly classified as responders, 21% correctly classified as non-responders (per-patient 3D set)

[81]

ML

Radiomics

Poor (high standard quality computed tomography scan, HQCT, set)

Poor (high standard quality computed tomography scan, SQCT, set)

Failed (HQCT set)

Failed (SQCT set)

[87]

ML

74 genes

97.6% correctly classified as responders, 100% correctly classified as non-responders

[95]

ML

16 long noncoding RNA

Good

[94]

ML

16 long noncoding RNA

Poor

[94]

ML

16 long noncoding RNA

Poor

[94]

Targeted therapy

DL

Radiomics

Good, 91.7% correctly classified as responders, 75% correctly classified as non-responders (four features model)

Good, 91.9% correctly classified as responders, 75% correctly classified as non-responders (four features model)

[98]

DL

Radiomics

Good, 97% correctly classified as responders, 59% correctly classified as non-responders

Good, 98% correctly classified as responders, 54% correctly classified as non-responders

[86]

ML

Radiomics

Good, 77% correctly classified as responders, 85% correctly classified as non-responders (high quality, HQ, cohort)

Good (standard quality, SQ, cohort)

Good, 80% correctly classified as responders, 78% correctly classified as non-responders (HQ cohort)

Poor, 82% correctly classified as responders, 61% correctly classified as non-responders (SQ cohort)

[87]

ML

Radiomics

89% correctly classified as responders, 85% correctly classified as non-responders (lesion model)

90% correctly classified as responders, 42% correctly classified as non-responders (lesion model)

92% correctly classified as responders, 86% correctly classified as non-responders (patient model)

[74]

ML

16-long noncoding RNA

Poor

[94]

ML

16-long noncoding RNA

Worthless

[94]

ML

16-long noncoding RNA

Poor

[94]

ML

67-gene

Good

Poor

[99]

Chemotherapy and targeted therapy

ML/DL

Imaging/molecular signatures

Good

Good

Present meta-analysis

Discussion

In this context, our meta-analysis demonstrated a good performance power of the learning models used to predict response to chemotherapy alone or combined to targeted therapy by discriminating response vs. non-response. Also, the calculation of overall HR shows that learning models have strong ability to predict improved survival. Next, when we have analyzed performances, both radiomics and molecular signatures showed to be effective to predict the response to therapy with good or excellent values. In particular, we found that both the delta radiomics and the 74 gene signatures were able to discriminate response vs. non-response by correctly classifying up to 99% of mCRC patients as responders and up to 100% of patients as non-responders.

(at 5° paragraph)

Conclusions

In addition, the delta-radiomics and the 74 gene signatures were found to able to discriminate response vs. non-response by correctly identifying up to 99% of mCRC patients as responders and up to 100% of patients as non-responders.

Abstract

Lastly, the delta-radiomics and the 74 gene signatures were able to discriminate response vs. non-response by correctly identifying up to 99% of mCRC patients as responders and up to 100% of patients as non-responders. Specifically, when we evaluated the predictive models with tests reaching the 80% sensitivity (SE) and the 90% specificity (SP), the delta radiomics showed the SE of 99% and the SP of 94% in the training set and the SE of 85% and the SP of 92 in the test set, whereas the 74 gene signatures the SE of 97.6% and the SP of 100% in the training set.

Question N.2 because the AUR is low

Answer:  please see the answer to Question 10 of the first reviewer (already reported).